# Reconciling global mean and regional sea level change in projections and observations

Jinping Wang[1,2], John A. Church [3✉], Xuebin Zhang [2✉] & Xianyao Chen [4]

The ability of climate models to simulate 20th century global mean sea level (GMSL) and regional sea-level change has been demonstrated. However, the Intergovernmental Panel on Climate Change (IPCC) Fifth Assessment Report (AR5) and Special Report on the Ocean and Cryosphere in a Changing Climate (SROCC) sea-level projections have not been rigorously evaluated with observed GMSL and coastal sea level from a global network of tide gauges as the short overlapping period (2007–2018) and natural variability make the detection of trends and accelerations challenging. Here, we critically evaluate these projections with satellite and tide-gauge observations. The observed trends from GMSL and the regional weighted mean at tide-gauge stations confirm the projections under three Representative Concentration Pathway (RCP) scenarios within 90% confidence level during 2007–2018. The central values of the observed GMSL (1993–2018) and regional weighted mean (1970–2018) accelerations are larger than projections for RCP2.6 and lie between (or even above) those for RCP4.5 and RCP8.5 over 2007–2032, but are not yet statistically different from any scenario. While the confirmation of the projection trends gives us confidence in current understanding of near future sea-level change, it leaves open questions concerning late 21st century non-linear accelerations from ice-sheet contributions.

[1] Department of Oceanography, College of Oceanic and Atmospheric Sciences, Ocean University of China, Qingdao, China. [2] Centre for Southern Hemisphere Oceans Research (CSHOR), CSIRO Oceans and Atmosphere, Hobart, TAS, Australia. [3] Climate Change Research Centre, University of New South Wales, Sydney, NSW, Australia. [4] Frontiers Science Center for Deep Ocean Multispheres and Earth System, Key Laboratory of Physical Oceanography, Ocean University of China, Qingdao, China. ✉email: john.church@unsw.edu.au; xuebin.zhang@csiro.au

As an essential indicator of global climate change and ocean variability, sea level has been simulated by models during both historical and future periods. Reliable sea-level projections are also vital for coastal communities. The IPCC AR5[1] and SROCC[2] provide global and regional sea-level projections, including estimates of contributions from oceans, glaciers, ice sheets, and land water from 2007 to 2100. The regional projections also include an allowance for relative sea level induced by glacial isostatic adjustment (GIA). Critically evaluating sea-level projections by comparing them with observations is important in enhancing our understanding of confidence in sea-level changes in the 21st century, identifying potential limitations in current projections, which helps to further calibrate and reduce uncertainties in the projections.

Previous studies have demonstrated the improved ability of models in simulating 20th century sea-level changes at both global and regional scales[3–5]. After 1950 and particularly for the satellite era since 1993, the model simulations accounted for essentially all the observed GMSL rise, with GMSL rise since 1970 dominated by anthropogenic climate change[6]. However, critically evaluating the sea-level projections by comparison with recent observations is challenging on both global and regional scales because (i) there is considerable natural variability and the overlapping period is short (2007–2018), and (ii) regional tide-gauge records are highly influenced by local factors, such as the vertical land motion (VLM)[7] and sea-level extreme events (like storm surges). The natural variability in sea-level change (e.g. the El Niño—Southern Oscillation (ENSO)[8]) and especially decadal (multidecadal) variations make the detection of trends and accelerations more difficult, even in long sea-level records[9].

Here, we show the trends of the IPCC AR5 and SROCC sea-level projections under three RCP scenarios from both GMSL and regional weighted mean at 177 tide-gauges stations agree well with satellite and tide-gauge observations over the common peiod 2007–2018 within 90% confidence level (90% CL), after considering the impacts of natural climate variability and correcting local residual VLM. Because of natural variability, we extend the period of observations and projections for a robust derivation of acceleration. We find the central values of observed GMSL (satellite altimeter over 1993–2018; sea-level reconstruction over 1970–2018) and regional weighted mean at tide-gauge stations (1970–2018) show larger accelerations than that from projections under RCP2.6 and lie between projected accelerations under RCP4.5 and RCP8.5, while not yet statistically different from any scenario. In the real word, the sea-level acceleration need to be reduced to be consistent with the lower and falling RCP2.6 mitigation emission scenario and the Paris targets in the late 21st century.

## Results

**Outline**. For GMSL, we utilise satellite altimeter observation time series over 1993–2018 from the Australian Commonwealth Scientific and Industrial Research Organization (CSIRO) and the U.S. National Aeronautics and Space Administration (NASA) Goddard Space Flight Centre (GSFC; Fig. 1a). Both groups attempted to correct instrumental drifts during the earlier altimeter period, by either using tide-gauge records with VLM estimated based on GIA or Global Positioning System (GPS) for CSIRO[10] or turning off the on-board calibration mode for GSFC[11]. We also use the long-term GMSL reconstructions based on tide-gauge records from Church & White 2011[12] and Dangendorf 2019[13] (hereafter referred to as CW2011 and D2019, respectively) over 1970–2018. Other GMSL reconstructions are not included here, as their differences from the above two reconstructions are generally small after 1970[14], e.g. GMSL from

ref. [15] closely resemble D2019 (Supplementary Fig. 1). For regional sea-level observations, 177 tide gauges around the world (Fig. 2) since 1970 are used.

To reduce the impact of natural climate variability, we use a multiple variable linear regression (MVLR) model[8] to represent the observed sea-level time series. The MVLR includes linear and quadratic changes with time and a linear dependence on several climate indices (Methods). The climate indices (Supplementary Fig. 2) are used to quantify the ENSO, the Pacific Decadal Oscillation (PDO), the Indian Ocean Dipole (IOD), the Southern Annular Mode (SAM), and the North Atlantic Oscillation (NAO). The low-frequency Atlantic Multidecadal Oscillation (AMO) and Atlantic Meridional Overturning Circulation (AMOC) indexes are not included in the MVLR model for the Atlantic Ocean because there may be a significant contribution of anthropogenic climate change in the recent AMO/AMOC changes[16–18]. Also, including the AMO/AMOC does not improve the skill of MVLR model in the Atlantic (more detailed discussion in Methods). The MVLR is applied to the altimeter GMSL over 1993–2018, and GMSL reconstructions and tide-gauge records over 1970–2018. The longer period (49 years), during which anthropogenic climate change dominates the sea-level rise (since 1970)[6], is required for robust regression of regional sea-level on low-frequency natural variability[19]. Furthermore, the uncertainty of the regional trend and acceleration estimates using the MVLR model during this period are substantially smaller than those derived over the short overlapping period of 2007–2018 (Methods; Supplementary Fig. 3).

The multimodel ensemble-mean outputs are evaluated in sea-level projections under three RCPs: 2.6, 4.5, and 8.5 for mitigation, medium and high emission scenarios, respectively[20]. For sea-level projections, the trend and acceleration are estimated directly by calculating linear and quadratic coefficients without the MVLR model, as the natural variability has been much reduced by multiple-model ensemble averaging. However, the procedure of multiple-model ensemble averaging in the production of the IPCC AR5 (as well as SROCC) projections reduces but does not completely eliminate the natural variability because of the limited ensemble size used for the AR5 projections (the RCP2.6 contains 16 models, and the RCP4.5 and 8.5 are based on 21 models)[21]. Therefore, we also use the low-pass filtered AR5 projections developed by CSIRO[22,23], in which the dynamic sea level (DSL), the only component directly simulated by the Coupled Model Intercomparison Project Phase 5 (CMIP5) models, is smoothed using a 20-year running-mean filter before adding to other sea-level contributions and computing the ensemble average. The DSL is the sea level relative to the geoid that is determined by the dynamical process associated with ocean density and circulation[24], while total sea level also includes the changes in the mass of ocean associated with glacier and ice-sheet mass loss or terrestrial water storage (TWS) changes. The low-pass filtered AR5 projections (AR5_lp hereafter) combine the GMSL contributions from the IPCC AR5 projections and are very similar to the AR5 projections, but the interannual to decadal natural variability has been much reduced, which helps us to focus on the climate change signals (Methods; Supplementary Fig. 4). The sea-level projections under three different RCPs have not diverged significantly from each other during our research period (2007–2032), but different accelerations lead to larger sea-level differences by 2100 (Supplementary Fig. 5)[1]. We use projections over the first 26 years 2007–2032 (the same length as the satellite data over 1993–2018) to evaluate the modelled global mean and regional sea-level accelerations. During this period, the projected accelerations start to be significantly different from zero on regional scales[9], and the uncertainty of the acceleration estimates is reduced by half comparing with the short overlapping period (Methods; Supplementary Fig. 6).

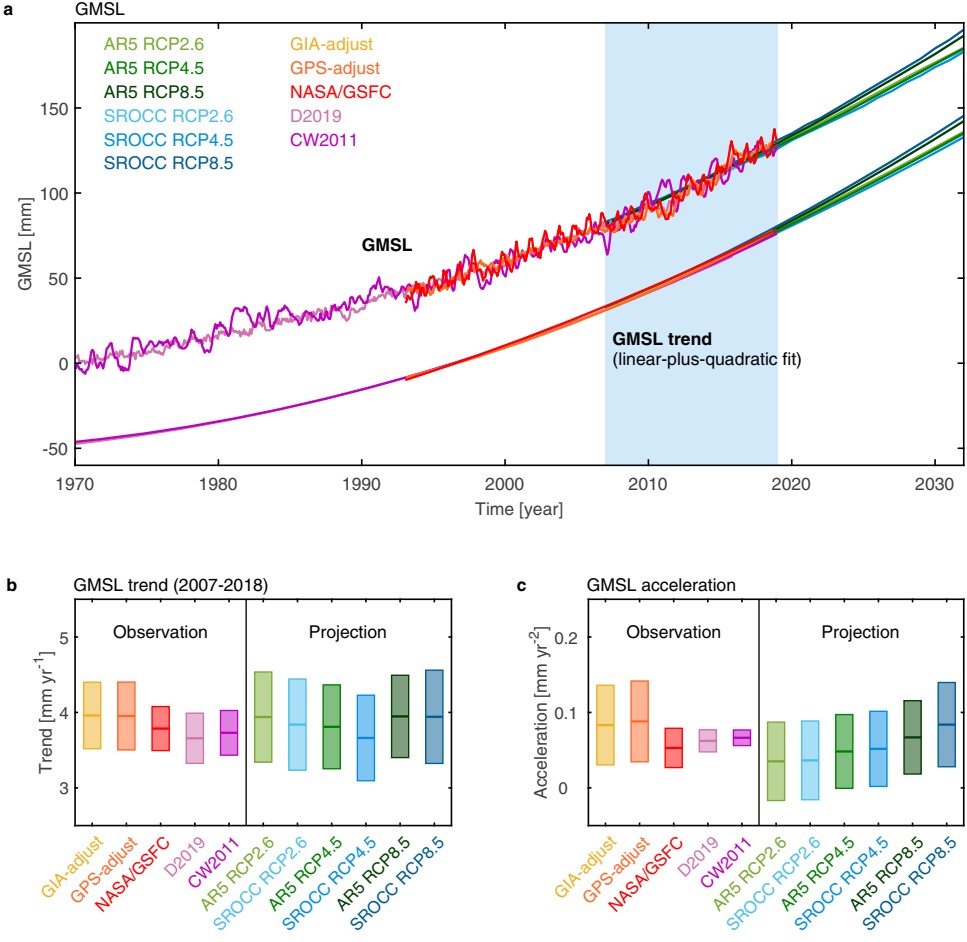

**Fig. 1 The global-mean sea level (GMSL) from observations compared with projections. a** Monthly satellite altimeter observations (1993–2018) with instrumental drifts corrected are from the Australian Commonwealth Scientific and Industrial Research Organization (CSIRO) based on glacial isostatic adjustment (GIA-adjust; yellow) and Global Positioning System (GPS-adjust; orange), as well as from the National Aeronautics and Space Administration Goddard Space Flight Centre (NASA/GSFC; red). The GMSL reconstructions (1970–2018) from ref. [13] (D2019; light purple) and ref. [12] (CW2011; purple) are smoothed with a 5-month running-mean filter. The annual multimodel averaged GMSL projections from the Intergovernmental Panel on Climate Change (IPCC) Fifth Assessment Report (AR5; from light green to dark green) and Special Report on the Ocean and Cryosphere in a Changing Climate (SROCC; from light blue to dark dark blue) under three Representative Concentration Pathway (RCP) scenarios respectively (2007–2032). GMSL trends including both linear and quadratic terms are also shown offset by −50 mm. The blue shaded area indicates the overlapping period between observations and projections (2007–2018). Box plots of **b** GMSL trends [mm yr⁻¹] over 2007–2018 and **c** GMSL acceleration [mm yr⁻²] over the whole period of each dataset. Error bars indicate 90% confidence level.

Note that the use of the MVLR means the trend is effectively evaluated during the overlapping period of the observations and projections, while the slowly varying anthropogenic acceleration is evaluated over a longer but not completely overlapping period to reduce the contamination from natural variability.

**GMSL trends**. The observed GMSL change contains considerable interannual variability, at least partly due to changes of TWS driven by ENSO[25,26], as well as decadal variability in TWS related to PDO[27,28], thermal expansion, and ice-sheet mass loss[29]. The underlying mechanisms between GMSL and other large-scale climate modes (e.g. AMO, IOD, and SAM) remain unclear. After minimising the natural variability related to ENSO and PDO via the MVLR (Methods Eq. [1]; Supplementary Fig. 7) from altimetry observations, the GMSL trend over 2007–2018 is $3.8 \pm 0.3$ mm yr⁻¹ for GSFC, or $4.0 \pm 0.4$ mm yr⁻¹ for CSIRO GIA-adjusted and GPS-adjusted data (90% CL; Table [1]), consistent with trends estimated from longer tide-gauge reconstructions (Table [1]). The observed trends during the overlapping period 2007–2018 (Table [1]) are all higher than estimates

from the whole altimeter records of about 3 mm yr⁻¹ since 1993[11,30,31] as a result of the accelerating rate of sea-level rise.

The GMSL trends estimated from AR5 projection for the common period 2007–2018 are almost identical to the trends estimated from observations over the same period (Table [1]), and certainly well within the 90% CL (Table [1]; Fig. 1b). The SROCC projections are the same as the AR5 projections except using an updated Antarctic ice-sheet dynamic contribution[2]. During our research period (2007–2032), the SROCC projections are virtually identical as the AR5 projections under all three scenarios (Table [1]; Supplementary Figs. 5d-f). Even at the end of 21st century, the SROCC projections are at most 10% different from the AR5 projections under the high emission scenario RCP8.5 (Supplementary Figs. 5a–c).

**GMSL accelerations**. The observed GMSL acceleration estimated over the satellite altimetry era (1993–2018) is $0.074 \pm 0.032$ mm yr⁻² from the GSFC altimeter time series ($0.109 \pm 0.060$ and $0.111 \pm 0.058$ mm yr⁻² for CSIRO GIA-adjusted and GPS-adjusted data, respectively) after removing the natural variability only related to

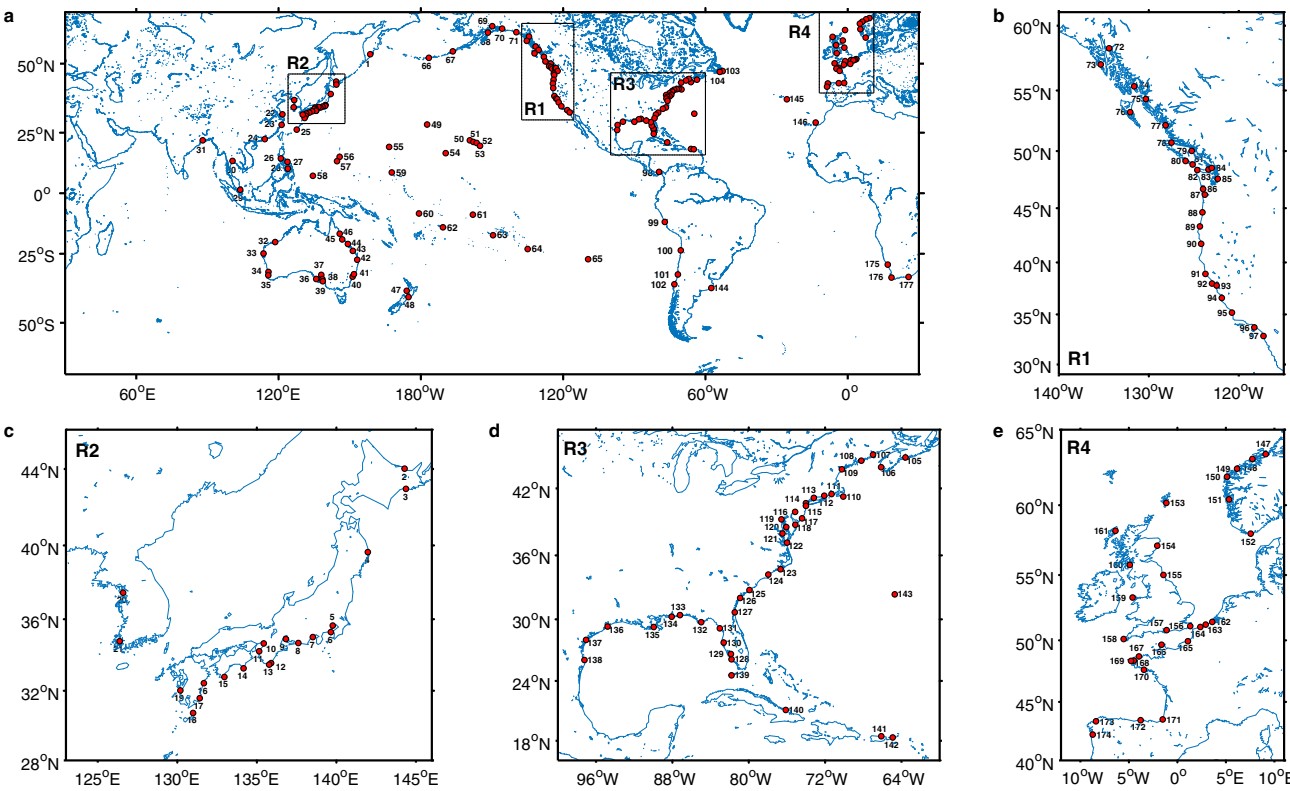

**Fig. 2 Locations of tide gauges. a** 177 tide-gauge (TG) stations used in this study (red circles). East coast of Pacific Ocean (EP) with TG identification (ID) number 66–102 including region R1 (**b**) contains 37 records. Asia nearby coastline (TG ID 1–31) contains 31 records, with an expanded region R2 (**c**) for details. Oceania (including Australia and New Zealand, TG ID 32–48) contains 17 records. The central Pacific Ocean (CP, TG ID 49–65) contains 17 records. West coast of Atlantic Ocean (WA, TG ID 103–144) including region R3 (**d**) contains 42 records. East coast of Atlantic Ocean (EA, TG ID 145–177) including region R4 (**e**) contains 33 records.

ENSO (Methods Eq. 2). Correcting for the recovery from the Mt Pinatubo eruption[32] would increase our GSFC estimate to $0.094 \pm 0.036 \, \mathrm{mm \, yr^{-2}}$ (90% CL), consistent with the previous estimate of $0.084 \pm 0.025 \, \mathrm{mm \, yr^{-2}}$ (one standard deviation) over 1993–2017 from ref. [33] with ENSO natural variability removed. After correcting for recovery from the Mt Pinatubo eruption, ref. [30] and ref. [31] estimated accelerations of $0.12 \, \mathrm{mm \, yr^{-2}}$ and $0.14 \pm 0.07 \, \mathrm{mm \, yr^{-2}}$ (90% CL). These would be reduced by about $0.03 \, \mathrm{mm \, yr^{-2}}$ if the ENSO variability was removed[33], consistent with our estimate of $0.094 \pm 0.036 \, \mathrm{mm \, yr^{-2}}$ within 90% CL. The GSFC acceleration reduces to $0.053 \pm 0.026 \, \mathrm{mm \, yr^{-2}}$ (or $0.083 \pm 0.053$ and $0.089 \pm 0.054 \, \mathrm{mm \, yr^{-2}}$ for CSIRO GIA-adjusted and GPS-adjusted data, respectively; Table 1) when both ENSO and PDO are removed via MVLR model (Methods Eq. 1; Supplementary Fig. 7), in agreement with accelerations estimated from the longer (1970–2018) GMSL reconstructions (e.g. $0.066 \pm 0.011 \, \mathrm{mm \, yr^{-2}}$ from CW2011; Fig. 1c; Table 1) within 90% CL. This indicates the PDO decadal variability potentially accounts for a small fraction of the acceleration in GMSL over the altimeter period. This corroborates the findings of previous studies[27,28], indicating that the decadal variability in the satellite GMSL records related to changes in TWS driven by PDO might obscure the anthropogenic acceleration of GMSL.

The AR5 GMSL accelerations over the first 26 years (2007–2032) are $0.035 \pm 0.052$, $0.048 \pm 0.048$, and $0.067 \pm 0.049 \, \mathrm{mm \, yr^{-2}}$ under RCP2.6, 4.5, and 8.5, respectively (Fig. 1c; Table 1). The SROCC projections indicate statistically equal accelerations with the AR5 projections under three RCP scenarios, with a slightly higher central value of acceleration (about 25%) than that from AR5 estimation under RCP8.5. This is associated with the larger contribution from Antarctic ice-sheet

dynamic response included in the SROCC[2]. The non-linear acceleration of Antarctic dynamic ice-sheet contribution is an important uncertainty in simulating sea level after 2050. The central values of the historical observed accelerations (including the multidecadal estimates with the smallest uncertainties; Table 1) are above the RCP2.6 accelerations and lie between (or above) the RCP4.5 and RCP8.5 accelerations from AR5 and SROCC projections (Table 1; Fig. 1c; Supplementary Fig. 8), but are not yet statistically different from any scenario. The projected GMSL accelerations are systemtically different in the future as a result of the different emission scenarios, e.g. the GMSL acceleration is declining and becomes negative after 2052 under RCP2.6 but is increasing under RCP8.5 (Supplementary Fig. 8b). Hence, there is a growing difference in the accelerations between the RCP scenarios in the future. The observed accelerations lie between (or above) the projected accelerations from RCP4.5 and RCP8.5 over 2007–2032 in the near future but will need to be reduced to be consistent with the lower and falling RCP2.6 acceleration which is closer to the Paris targets.

**Regional sea-level rise and contributions from natural variability.** Regional sea level exhibits considerably more natural variability than the GMSL (Fig. 3)[19], implying the need to consider longer analysis periods. We focus on evaluating the projections of coastal sea level as recorded by tide gauges and used in coastal management (rather than offshore observations from satellites), after minimising the natural variability by using the MVLR analysis (Methods Eqs. 3, 4). We also applied the MVLR analysis to the regional ocean-reanalysis data (Ocean ReAnalysis System 5, ORAS5) over the same period as tide-gauge

**Table 1 Comparison of observed trends and accelerations with those from projections.**

|  | Trend [mm yr$^{-1}$] | Acceleration [mm yr$^{-2}$] |
|---|---|---|
| Altimeter GMSL | 2007–2018 | 1993–2018 |
| GIA-adjust | 4.0 ± 0.4 | 0.083 ± 0.053 |
| GPS-adjust | 4.0 ± 0.4 | 0.089 ± 0.054 |
| NASA/GSFC | 3.8 ± 0.3 | 0.053 ± 0.026 |
| GMSL reconstruction | 2007–2018 | 1970–2018 |
| D2019 | 3.7 ± 0.3 | 0.062 ± 0.014 |
| CW2011 | 3.7 ± 0.5 | 0.066 ± 0.011 |
| AR5 projections | 2007–2018 | 2007–2032 |
| RCP2.6 | 3.9 ± 0.6 | 0.035 ± 0.052 |
| RCP4.5 | 3.8 ± 0.6 | 0.048 ± 0.048 |
| RCP8.5 | 3.9 ± 0.5 | 0.067 ± 0.049 |
| SROCC projections | 2007–2018 | 2007–2032 |
| RCP2.6 | 3.8 ± 0.6 | 0.036 ± 0.052 |
| RCP4.5 | 3.7 ± 0.6 | 0.052 ± 0.050 |
| RCP8.5 | 3.9 ± 0.6 | 0.084 ± 0.056 |
| TG weighted mean at TG locations | 2007–2018 | 1970–2018 |
| TG | 3.6 ± 1.7 | 0.063 ± 0.120 |
| AR5 regional weighted mean at TG locations | 2007–2018 | 2007–2032 |
| RCP2.6 | 4.0 ± 1.3 | 0.021 ± 0.085 |
| RCP4.5 | 3.7 ± 0.9 | 0.053 ± 0.063 |
| RCP8.5 | 3.9 ± 0.8 | 0.073 ± 0.088 |
| SROCC regional weighted mean at TG locations | 2007–2018 | 2007–2032 |
| RCP2.6 | 3.9 ± 1.3 | 0.020 ± 0.085 |
| RCP4.5 | 3.6 ± 0.9 | 0.056 ± 0.063 |
| RCP8.5 | 3.8 ± 0.8 | 0.089 ± 0.087 |
| AR5_lp regional weighted mean at TG locations | 2007–2018 | 2007–2032 |
| RCP2.6 | 4.1 ± 1.3 | 0.041 ± 0.022 |
| RCP4.5 | 3.9 ± 1.3 | 0.053 ± 0.019 |
| RCP8.5 | 4.1 ± 1.3 | 0.072 ± 0.024 |

Global mean and regional weighted-mean sea-level trends [mm yr$^{-1}$] at tide-gauge (TG) stations over the common period (2007–2018) and accelerations [mm yr$^{-2}$] during the whole study period of each data. Altimeter and reconstructed global-mean sea level (GMSL) have the climate variability related to the El Niño—Southern Oscillation (ENSO) and Pacific Decadal Oscillation (PDO) removed over the whole period of the records. The uncertainties represent the 90% confidence level. The satellite altimeter observations with instrumental drifts corrected are from the Australian Commonwealth Scientific and Industrial Research Organization (CSIRO) based on glacial isostatic adjustment (GIA-adjust) and Global Positioning System (GPS-adjust), as well as from the National Aeronautics and Space Administration Goddard Space Flight Centre (NASA/GSFC). The GMSL reconstructions are from ref. [13] (D2019) and ref. [12] (CW2011). The sea-level projections are including the Intergovernmental Panel on Climate Change (IPCC) Fifth Assessment Report (AR5), Special Report on the Ocean and Cryosphere in a Changing Climate (SROCC) and the low-pass filtered AR5 projections (AR5_lp) under three Representative Concentration Pathway (RCP) scenarios.

observations (1970–2018) to provide large-scale regional patterns. The MVLR analysis was not applied to regional satellite altimeter observations because the 26-year study period is still quite short to reasonably separate the regional sea-level trends and accelerations from low-frequency decadal variability (Methods; Supplementary Fig. 3).

In the Indo-Pacific Ocean, the regional sea-level variability is dominated by the ENSO (Fig. 3a) and PDO (Fig. 3b) over 1970–2018 based on the ORAS5 reanalysis, and the patterns are similar to those shown in previous studies[8,19], with significant east-west seesaw pattern of sea level across the tropical to subtropical Pacific Ocean. The IOD and SAM contributions are much smaller (Fig. 3d, f). The NAO related sea level (Fig. 3c) resembles the dominant tripole pattern of sea-level variability in the North Atlantic during the satellite altimetry period[34]. The goodness of fit of the MVLR analyse can be evaluated by the ratio of variance explained by the regression over the total variance of sea level at each location (R$^2$; Supplementary Fig. 9). Together the high-pass filtered ENSO and low-pass filtered PDO have a mean R$^2$ of 15% among all tide gauges in Indo-Pacific Ocean, increasing regression skill in the tropical and subtropical Pacific Ocean (Supplementary Fig. 9c). The average percentage of sea-level variance explained by trend and quadratic terms and all climate indices over all tide gauges is 45% (Supplementary Fig. 9f).

After removing the variability associated with climate modes using the MVLR, the linear trend map is positive almost everywhere (Fig. 3e), except small regions in the Southern Ocean and North Atlantic subpolar gyre in the ORAS5 reanalysis. The region with the largest negative trend (< −4 mm yr$^{-1}$) in the Southern Ocean (in the Pacific sector, centred at 150ºW) may be associated with the delayed ocean warming due to circumpolar upwelling and equatorward heat transport driven by Southern Ocean's meridional overturning circulation[35]. The sea-level acceleration is evident at most of the tide gauges and in most areas around the world, with a notable positive band in the mid-to-high latitudes in the Southern Hemisphere (Fig. 3g). The dominance of extratropical Southern Hemisphere acceleration is consistent with the stronger warming during the Argo era 2006–2013[36]. This region has larger signal-to-noise ratio making for easier early detection of the anthropogenic signal[21], suggesting long-term climate change signals may be starting to emerge in this region[37].

**Residual VLM contribution to local sea-level change**. Tide gauges are located along the coasts and may be affected by crustal movement on small local spatial scales. Currently land subsidence or uplift rate exceeds the sea-level rate at several coastal cities[7]. The AR5 sea-level projections are produced as sea level relative to the sea floor (relative sea level, RSL) by including simulated GIA associated with the last deglaciation and contemporary GRD (changes in Earth Gravity, Earth Rotation and viscoelastic solid-Earth Deformation)[24] fingerprint related to present-day mass redistribution (e.g. contemporary polar ice sheets melting)[1]. However, they do not include local factors like tectonics, sediment compaction and anthropogenic subsidence, which may also contribute to RSL changes at tide-gauge locations. Because these local factors are not included in the IPCC projections, we use the method of ref. [14] to separate the total VLM into three components related to GIA, contemporary GRD, and a residual VLM term (Methods Eq. 5; Fig. 4), with the purpose to better compare tide-gauge observations with sea-level projections. The residual VLM correction is then applied to the 177 tide-gauge records, so that the sea-level projections can be evaluated fairly. The residual VLM indicates large spatial variations (Fig. 4; Supplementary Fig. 10), e.g., the strong subsidence rate (<−10 mm yr$^{-1}$) in Manila (TG ID 26) is related to anthropogenic groundwater depletion[38]; The uplift of Alaska coast (~15 mm yr$^{-1}$ at TG ID 71–72) results predominantly from isostatic rebound associated with glacier mass loss over the post-Little Ice Age period[39].

**Regional sea-level trends**. A direct comparison of regional trends between tide-gauge observations and the AR5 projection over 2007–2018 (Fig. 5 top panel) indicates a number of significant local discrepancies. The histogram of difference between tide-gauge and AR5 shows substantially skewed non-Gaussian distribution, with a long tail of negative values due to the significant residual VLM over relatively small spatial scales that were not included in the AR5 projections (Fig. 4).

When the estimated residual VLM corrections (Methods) were applied to the tide-gauge records, the Root Mean Square Difference (RMSD) between observed and projected sea-level trends decreases from 3.9 to 2.0 mm yr$^{-1}$ (Fig. 5 middle panel), and further to 1.7 mm yr$^{-1}$ after also removing climate variability

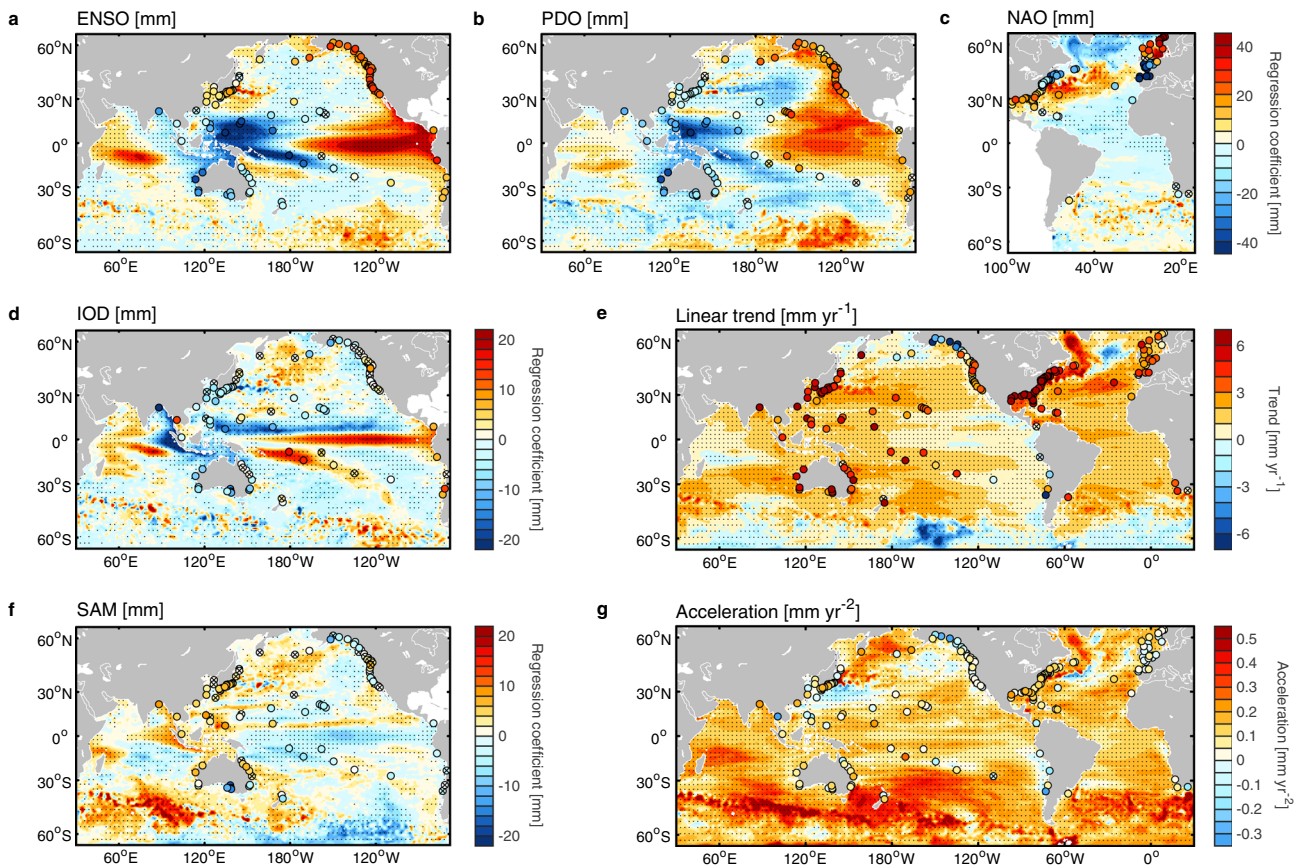

**Fig. 3 Regression coefficient maps from the multiple variable linear regression (MVLR) analysis.** Colored circles indicate significant regression coefficients (90% confidence level) of tide-gauge data (1970–2018) related to **a** the El Niño—Southern Oscillation (ENSO), **b** the Pacific Decadal Oscillation (PDO), **c** the North Atlantic Oscillation (NAO) (all with the same colour scale), **d** the Indian Ocean Dipole (IOD), **e** linear trend, **f** the Southern Annular Mode (SAM), and **g** acceleration (twice the quadratic term). Regression coefficients of tide-gauge records, which are not significant at 90% confidence level are denoted as cross symbols (hereafter). The corresponding coefficient contours are based on the Ocean Reanalysis System 5 (ORAS5) ocean-reanalysis over the same period. Stippling indicates where the coefficients are statistically significant at the 90% confidence level for ORAS5 reanalysis.

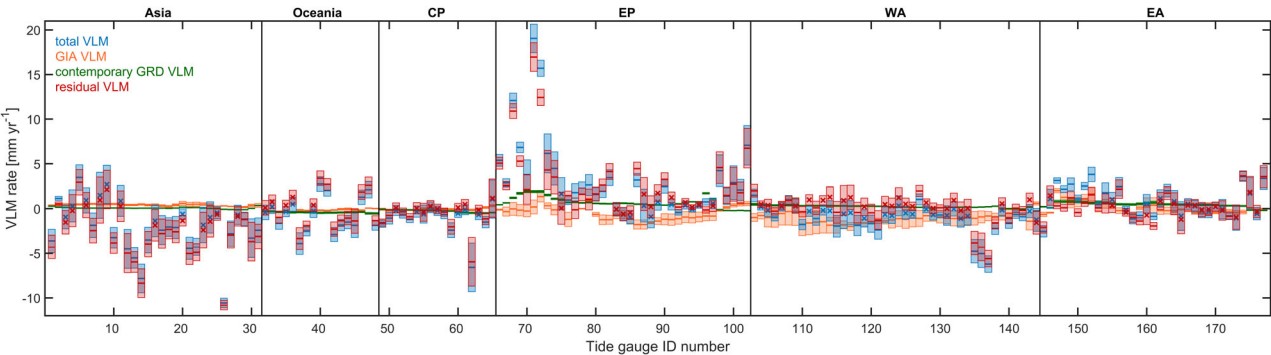

**Fig. 4 Estimated rates of total vertical land motion (VLM) and different components at tide-gauge stations.** Box plots of total VLM (blue), glacial isostatic adjustment (GIA) VLM (green), contemporary changes in Earth Gravity, Earth Rotation and viscoelastic solid-Earth Deformation (GRD) VLM (orange), and residual VLM (red) rates. Negative (positive) values denote subsidence (uplift). Error bars indicate 90% confidence level, and trends, which are not significant at 90% confidence level denoted as cross symbols. Region definition is shown in Fig. 2.

via MVLR (Fig. 5 bottom panel). The weighted mean (by 1/uncertainty squared and hereafter) of all tide-gauge trends (3.6 ± 1.7 mm yr$^{-1}$; red horizontal line in Fig. 5b bottom panel) is statistically equivalent to the weighted-mean trends from the AR5 projections under RCP4.5 (3.7 ± 0.9 mm yr$^{-1}$; blue horizontal line in Fig. 5b bottom panel) over 2007–2018, with their residual

trend histogram (green in Fig. 5b bottom panel) having a mean value nearly at zero. Furthermore, the residual in the trend shows that the tide-gauge observations systematically reveal slightly (but not significantly) higher trends than model projections along the North America East Coast (Fig. 5a bottom panel, TG ID 107–134), while a little lower than projected trend along the

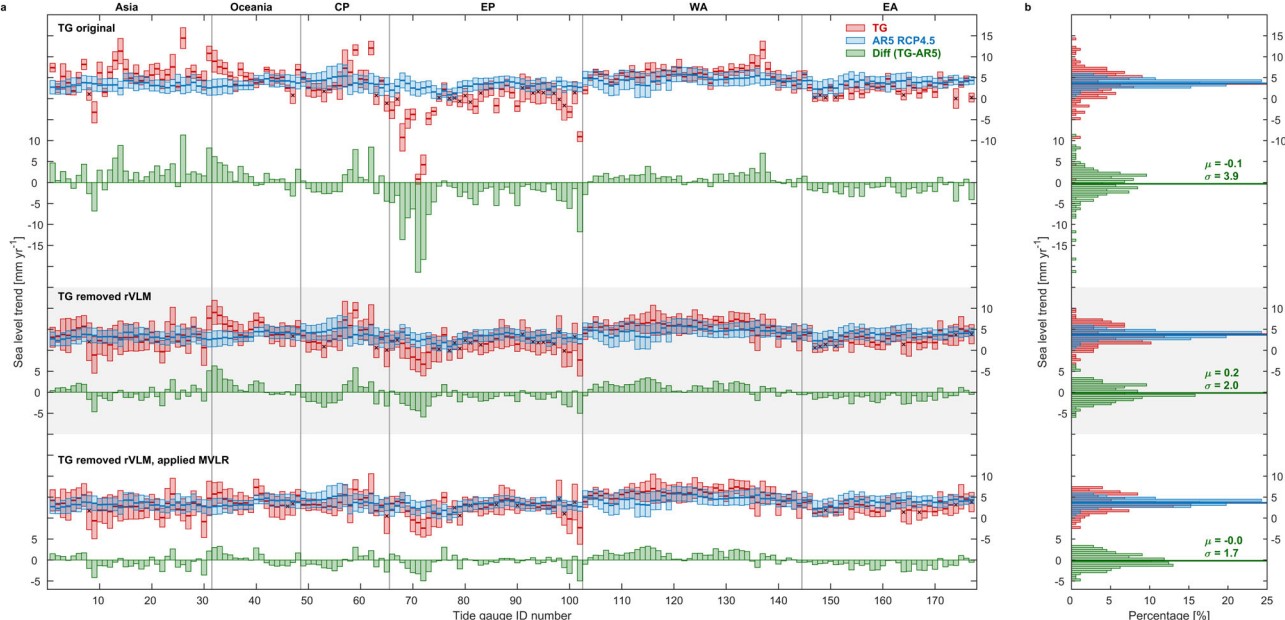

**Fig. 5 Regional sea-level trends (2007–2018) from tide-gauge observations compared with the sea-level projection.** Sea-level trends [mm yr$^{-1}$] are based on tide-gauge observations (TG; red), the Intergovernmental Panel on Climate Change (IPCC) Fifth Assessment Report (AR5) projection under Representative Concentration Pathway (RCP) 4.5 scenario (blue) and their difference (TG minus AR5; green) at each tide-gauge station, for **a** box plots and **b** histogram. Error bars indicate 90% confidence level, and trends which are not significant at 90% confidence level denoted as cross symbols. Region definition in **a** is shown in Fig. 2. TG trends have no adjustment in top panel, residual vertical land motion (rVLM) adjustment in middle panel, and rVLM adjustment and climate variability removed via multiple variable linear regression (MVLR) model in bottom panel. In histograms **b**, the bin width is 0.5 mm yr$^{-1}$, horizontal lines present weighted-mean trend ($\mu$) at all TG stations, $\sigma$ denotes the standard deviation of the trend at all TG stations.

European Coast (Fig. 5a bottom panel, TG ID 151–174). This could be because only the dominant basin-scale natural variability (i.e. NAO) is considered here and other natural variabilities especially on sub-basin scale have not been adequately considered, or because biases may still exist in VLM and/or the regional fingerprints of recent mass loss used in the projections. Regional trends from AR5 RCPs 2.6 and 8.5 as well as three scenarios from SROCC and AR5_lp projections all agree with the observed sea-level rise well within the 90% CL (Supplementary Figs. 11–13).

The AR5, SROCC, and AR5_lp projections indicate sea-level rise almost everywhere during 2007–2018, but the AR5_lp projections are temporally smoother and have greater similarities among the three scenarios because of the more effective removal of the natural variability in AR5_lp projections (Supplementary Fig. 14). Compared with tide-gauge trends after correcting residual VLM and removing climate variability, AR5 projections show slightly higher trend pattern along the Alaska coast. This may be due to inaccuracies in the GIA in this region (global GIA model usually does not include the local low-viscosity upper mantle here[40,41]), in the GRD fingerprints with Alaskan glacier loss or in the regional dynamical sea-level projections.

**Regional sea-level accelerations.** The majority (80%) of 177 tide-gauge records used in this study have a positive sea-level acceleration since 1970 (Fig. 6). Only four locations have significant deceleration (less than −0.2 mm yr$^{-2}$). The deceleration at TG ID 30 (Thailand) is likely related to a recent decrease of groundwater pumping[42], and the decelerations at TG ID 70, 71, and 102 are likely the result of the nearby accelerating glacier melting measured by Gravity Recovery and Climate Experiment (GRACE) satellite gravimetry[43]. The regional sea-level projections are not able to capture these very local decelerations, therefore it is

challenging to reconcile with the current generation of regional sea-level projections based on coarse-resolution climate model simulation.

The weighted mean of the observed accelerations over all gauges (0.063 ± 0.120 mm yr$^{-2}$) has a central value lying between the projected accelerations under RCP4.5 (0.053 ± 0.063 mm yr$^{-2}$) and RCP8.5 (0.073 ± 0.088 mm yr$^{-2}$). The projected sea-level accelerations in many stations under RCP2.6 from AR5 projections are close to zero and not significant, with a lower weighted-mean value of 0.021 ± 0.085 mm yr$^{-2}$ (Fig. 6 top panel). Regional accelerations estimated by SROCC projections (Supplementary Fig. 15) are generally the same as the AR5 projections under all three scenarios, while the regional accelerations from AR5_lp at the tide gauges (Supplementary Fig. 16) have narrower histograms than the AR5 projections (Fig. 6). The regional weighted-mean AR5_lp accelerations are significantly differently from zero at 0.053 ± 0.019 mm yr$^{-2}$ for RCP4.5 (0.041 ± 0.022 mm yr$^{-2}$ for RCP2.6; 0.072 ± 0.024 mm yr$^{-2}$ for RCP8.5). That is because the AR5_lp projections produce a more uniform large-scale acceleration than the AR5 and SROCC projections due to the reduced natural variabilities (Supplementary Fig. 17). As a result, the AR5_lp sea-level projections under different scenarios can be more clearly distinguished by acceleration in the earlier decades of the 21st century than AR5 and SROCC projections.

## Discussion

Sea-level (oceans and ice sheets in particular) changes have long time scales and the overlapping period between observations and projections is short for evaluation purpose. Our study is a first attempt to evaluate both linear trend and acceleration of sea-level projections on global and regional scales complementing previous evaluation of the historical sea-level model simulation during the 20th century[3–5].

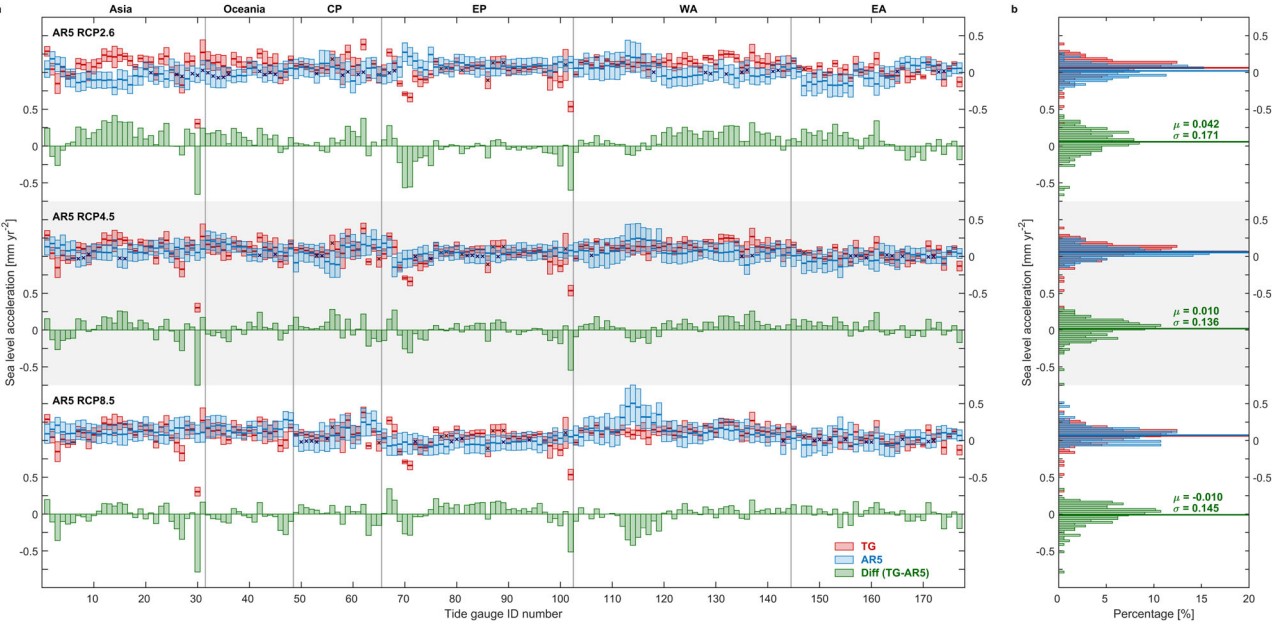

**Fig. 6 Regional sea-level accelerations from tide-gauge observations (1970–2018) compared with the sea-level projection (2007–2032).** Sea-level acceleration [mm yr$^{-2}$] are based on tide-gauge observations (TG; red), the Intergovernmental Panel on Climate Change (IPCC) Fifth Assessment Report (AR5) projection (blue) and their differences (TG minus AR5; green) at each tide-gauge station, for **a** box plots and **b** histogram, with AR5 under Representative Concentration Pathway (RCP) 2.6 (top panel), RCP4.5 (middle panel), and RCP8.5 (bottom panel) scenarios. TG accelerations with climate variability removed via multiple variable linear regression (MVLR). Error bars indicate 90% confidence level, and accelerations, which are not significant at 90% confidence level denoted as cross symbols. Region definition in **a** is shown in Fig. 2. In histograms **b**, the bin width is 0.025 mm yr$^{-2}$, horizontal lines present weighted-mean accelerations ($\mu$) at all TG stations, σ denotes the standard deviation of the acceleration at all TG stations.

After minimizing impacts of natural variabilities and correcting the local residual VLM, we find the projected trends from AR5, SROCC and AR5_lp projections are closely consistent with observations on both global and regional scales during the overlapping period 2007–2018 (Table 1). The differences between observed and projected sea-level trends are less than 0.5 mm yr$^{-1}$ for both global mean (Fig. 1) and weighted-mean regional sea-level trends (Fig. 5; Supplementary Figs. 11–13; well within the uncertainty bounds over the short comparison period), consistent with evaluations of sea-level models for the 20th century[3–5]. Our result contrasts with the finding of an average of 2 mm yr$^{-1}$ larger projections than observations at 19 tide-gauge locations in a recent study[44]. We can reproduce the results of that study if we low-pass filter the data sufficiently during the whole 20th century. However, the pronounced acceleration in the rate of sea-level rise during the second half of the 20th century[1,3] resulting from anthropogenic climate change[6] will be removed by this filtering, leading to an underestimation of recent rate of sea-level rise.

The observed GMSL accelerations from satellite altimeter (e.g. 0.053 ± 0.026 mm yr$^{-2}$ from GSFC, or 0.083 ± 0.053 mm yr$^{-2}$ from CSIRO GIA-adjusted and 0.089 ± 0.054 mm yr$^{-2}$ from CSIRO GPS-adjusted data) lie between (or even above) the RCP4.5 and 8.5 projections (Table 1), after removing impacts of the high-pass filtered ENSO and low-pass filtered PDO. The accelerations from the longer GMSL reconstructions have smaller uncertainties (0.062 ± 0.014 mm yr$^{-2}$ from D2019; 0.066 ± 0.011 mm yr$^{-2}$ from CW2011) and are again between the RCP4.5 and 8.5 projections and well above that for RCP2.6. The altimeter observed acceleration without removing PDO related variability (e.g. 0.074 ± 0.032 mm yr$^{-2}$ from GSFC) is larger than that from AR5 RCP8.5 scenario, which suggests that robust acceleration estimation requires improved understanding of low-frequency natural variability. The acceleration in the real world

will need to be reduced to be consistent with the lower and falling RCP2.6 acceleration and the Paris targets (Supplementary Fig. 8b).

On regional scales, the majority of 177 tide-gauge records (80%) have positive accelerations over 1970–2018, with their weighted-mean value (0.063 ± 0.120 mm yr$^{-2}$) also lying between the RCP4.5 and RCP8.5 projections, as for the GMSL acceleration. However, because of the short period available for comparison, the regional accelerations that are not yet statistically different to the projections for any of the scenarios. The AR5_lp projections indicate a significant regional weighted-mean accelerations (e.g. 0.053 ± 0.019 mm yr$^{-2}$ under RCP4.5), which highlights the importance of identifying and possibly removing the natural variability in detecting any anthropogenic acceleration in sea-level rise. The earlier detection of a significant increase in the rate of sea-level rise is important in informing the public and supporting direct adequate mitigation and adaptation responses. With the long timescale of ocean thermal expansion and ice-sheet contributions, the agreement between the observed and projected accelerations do not bode well for sea-level impacts over coming decades.

The IPCC AR5 (and later) process-based sea-level projections, evaluated rigorously with various observations like this study and with a continued focus to improve regional patterns, will be important for our understanding of future global and regional sea-level changes. This evaluation gives us confidence in our present understanding of sea-level changes and the sea-level projections for the next several decades provided by the IPCC but does not directly address open questions concerning future non-linear accelerations from ice-sheet contributions.

## Methods

**Satellite altimetry data.** Previous comparison of sea level between satellite altimetry and tide gauges suggests that the altimetry system has significant systematic

bias caused by instrumental drifts during the first 6 years (1993–1999) of the TOPEX-A altimeter operation[10], which can affect the estimated the trend and acceleration of observed sea level. A more recent study concluded that this drift is likely caused by using a "calibration mode" correction on the TOPEX-A data[11]. The NASA/GSFC product corrects this bias by not applying the "calibration mode" corrections to the original TOPEX-A data, yielding results consistent with those found by ref. [10]. In this study, we use three different altimetry-based monthly GMSL time series over 1993–2018 from two processing groups: CSIRO[10] and the NASA/GSFC[11]. All datasets were downloaded in April 2019. An altimeter measures geocentric sea level and does not detect solid Earth deformation. Therefore, to compare altimeter sea level with AR5 projections, we applied GIA and contemporary GRD corrections to the satellite data to account for the deformation of the solid Earth to estimate changes in ocean water volume (RSL). These corrections increased the altimetry GMSL linear trend by 0.30 mm yr$^{-1}$ (GIA correction)[45] and 0.13 mm yr$^{-1}$ (contemporary GRD correction)[46], respectively. The related contemporary GRD acceleration would be small (about 5% of the mass acceleration) and is not included here. We use error estimates of 0.06 mm yr$^{-1}$ for the GIA correction from ref. [47] and 0.01 mm yr$^{-1}$ for the contemporary GRD correction from ref. [46] (one standard deviation) and add these in quadrature to the uncertainties of GMSL trend. Then the 90% CL is computed as the one standard deviation (STD) uncertainty multiplied by 1.65, assuming a normal distribution.

We also use monthly sea-level fields of the combined TOPEX/Poseidon, Jason-1, Jason-2 and Jason-3 observations with the TOPEX-A bias corrected by CSIRO with spatial resolutions of 1º (updated in May 2019 by Benoit Legresy, with permission to use). The closest grid point to each tide-gauge station is selected to calculate VLM rates[7] (see below).

**Tide-gauge records**. We use monthly tide-gauge (TG) RLR (revised local reference) data from the Permanent Service for Mean Sea Level (PSMSL; data downloaded in January 2019)[48] around the world except polar regions (65ºS–65ºN and 0ºE–360ºW). Careful selection and editing criteria are used[49]. The stations with a sudden sea-level record jump of more than 500 millimetres between two consecutive months are excluded. Tide gauges are included only if the total gap length is shorter than 20 years during 1970–2018. We also remove tide gauges, which have missing data for more than 10 years at the beginning, or longer than 5 years at the end, as the estimate of quadratic term is sensitive to the beginning and end of the time series and trends are evaluated during the overlapping period (2007–2018). Stations located in semi-enclosed Mediterranean Seas and Lakes (the Red Sea, Persian Gulf, Black Sea, Caspian Sea, Sea of Japan, Hudson Bay, Great Lakes, Baltic Sea, and Mediterranean Sea) are not included in this study, since the sea-level projections in these regions are less reliable because low-resolution climate models tend to not represent semi-included seas well (e.g. see Mediterranean Sea in ref. [50]). 177 stations are available since 1970 after applying the above selection criteria (Fig. 2). Gaps in tide-gauge records are not filled. For all monthly data used in this study, the long-term mean seasonal cycle over the whole period is removed and a 5-month running-mean filter is applied before further analysis.

**Reanalysis data**. Monthly sea-level reanalysis data from the European Centre for Medium-Range Weather Forecast (ECMWF) ORAS5 (data downloaded in September 2018)[51] since 1970 is used in this study (the same length as tide-gauge records). We only use the ORAS5 reanalysis to provide complementary spatial information over the same period as tide-gauge observation (Fig. 3), but not for evaluating sea-level projections and the drawing main conclusion.

**Climate indices**. Six climate indices are used: the Multivariate ENSO Index[52], the PDO Index[53], the IOD represented by the Indian Ocean Dipole Mode[54], the SAM Index[55], and the NAO Index[56]. All climate indices were accessed in April 2019. The time series are smoothed following the ideas from refs. [8,19]. A 6-year low-pass Lanczos filtering[57] is applied to the PDO Index to better represent decadal to interdecadal variability. The high-pass filtered ENSO Index is derived by smoothing with a 5-month running mean and then subtracting the low-frequency component, which is estimated by 6-year Lanczos low-pass filtering of ENSO index. This high-pass filtered ENSO index mainly represents interannual and shorter timescale variability (Supplementary Fig. 2). The correlation between high-pass filtered ENSO and low-pass PDO is nearly zero (0.05), which suggests independence of them due to filtering. Lanczos filtering used here helps to keep the data at the boundary as the quadratic term estimation is sensitive to beginning and end of the time series. The IOD, SAM and NAO indices are smoothed with a 5-month running-mean filter. All indices are normalised by their STD.

**Multiple variable linear regression**. In order to focus on the observed sea-level trend and acceleration and separate them from the climate variability, we use the MVLR model[8] on observed GMSL time series (Eqs. 1, 2), each tide-gauge station and each grid point of ORAS5 reanalysis (Eqs. 3, 4) as follows.

For GMSL observations, we use the high-pass filtered ENSO (interannual and shorter time scales) and low-pass filtered PDO (decadal to interdecadal variability; Supplementary Fig. 2) to remove natural variability from the GMSL. The MVLR

between GMSL ($\widehat{SL}_G$) and the indices is:

$$\widehat{SL}_G = b_{G0} + b_{G1}t + b_{G2}t^2 + b_{G3}\mathrm{ENSO} + b_{G4}\mathrm{PDO} + \varepsilon_G \quad (1)$$

In all MVLR models (including Eqs. 1–4), $\widehat{SL}_*$ denotes the sea-level observations, $b_{*0}$ is the intercept, the coefficient $b_{*1}$ represents the sea-level linear trend, the quadratic coefficient $b_{*2}$ can be converted to an acceleration ($2 \times b_{*2}$), and $\varepsilon_*$ is the residual term. The subscript $G$ in Eq. (1) denotes the GMSL observations (over 1970–2018 for reconstructions and 1993–2018 for satellite observations), $SAT$ in Eq. (2) presents the altimeter GMSL records, $P$ in Eq. (3) is for the regional sea level in Indo-Pacific Ocean, and $A$ in Eq. (4) denotes the regional sea level in the Atlantic Ocean. In Eq. (1), the $b_{G3}$ and $b_{G4}$ are regression coefficients related to the ENSO and PDO index, respectively, for the GMSL. Although there are also other definitions of acceleration allowing us to detect the change of acceleration during a research period[58], the quadratic term estimated by the MVLR model is a robust method to estimate acceleration by using all the data and simultaneously allowing minimization of the natural variability signals. The objective of the MVLR analysis is to minimize natural climate variability on the trends $b_{*1}$ and accelerations $2 \times b_{*2}$. However, because our study period is still quite short and there may be other low-frequency natural variability not considered here, these coefficients could potentially still be affected by natural variability. After applying the MVLR to GMSL over 1993–2018 and tide-gauge records over 1970–2018, we derive accelerations to provide information on temporal changes in the sea-level rate, and the linear trends during the overlapping period (2007–2018), with the projections, with the natural variability minimised.

The explained variance ratio ($R^2$) can objectively measure the goodness of fit of the MVLR. The higher $R^2$ is, the better the regression is. The $R^2$ values of all GMSL observations are close to 1, which are dominated by the linear trend. The contributions from ENSO and PDO variability on GMSL are relatively small comparing with linear-plus-quadratic term for both altimeter records and tide-gauge reconstructions (Supplementary Fig. 7). Considering the relatively short altimeter period, we also apply the MVLR model on altimeter observed GMSL ($\widehat{SL}_{SAT}$) including only the unfiltered ENSO index (ENSO$_O$):

$$\widehat{SL}_{SAT} = b_{SAT0} + b_{SAT1}t + b_{SAT2}t^2 + b_{SAT3}\mathrm{ENSO}_O + \varepsilon_{SAT} \quad (2)$$

where $b_{SAT3}$ is the regression coefficient related to the unfiltered ENSO index (ENSO$_O$) for the satellite observations.

On regional scales, the PDO, ENSO, IOD, SAM, and NAO indices are used in the MVLR analysis to minimise the impact of natural variability on the tide-gauge data. For the Indo-Pacific Ocean sea levels ($\widehat{SL}_P$), four climate indices (ENSO, PDO, IOD, and SAM) are used in the MVLR model following ref. [19]:

$$\widehat{SL}_P = b_{P0} + b_{P1}t + b_{P2}t^2 + b_{P3}\mathrm{ENSO} + b_{P4}\mathrm{PDO} + b_{P5}\mathrm{IOD} + b_{P6}\mathrm{SAM} + \varepsilon_P \quad (3)$$

where the $b_{P3}$ to $b_{P6}$ are the regression coefficients related to the high-pass filtered ENSO, low-pass filtered PDO, IOD, and SAM indices, respectively, in the Indo-Pacific Ocean.

For the Atlantic Ocean sea levels ($\widehat{SL}_A$), the NAO index is used in MVLR model:

$$\widehat{SL}_A = b_{A0} + b_{A1}t + b_{A2}t^2 + b_{A3}\mathrm{NAO} + \varepsilon_A \quad (4)$$

where $b_{A3}$ is the regression coefficient related to the NAO index in the Atlantic Ocean.

For the linear trend ($b_{*1}$), $R^2$ values are high at most stations along the North America East Coast (Supplementary Fig. 9a). Some tide gauges located along Japan, Alaska and Gulf of Mexico coasts show extremely high $R^2$ values (close to 1), which are dominated by the linear trend. $R^2$ values of acceleration ($b_{*2}$) are generally low at most stations around the world (Supplementary Fig. 9b). Adding ENSO and PDO indices increases regression skill in tropical and subtropical Pacific Ocean (Supplementary Fig. 9c).

We attempted to remove the low-frequency AMO/AMOC signal from the regional sea level in the Atlantic Ocean. The AMOC index[59] related sea-level pattern has higher sea level in the subpolar gyre in the North Atlantic Ocean and lower sea level near the Gulf Stream path (Supplementary Fig. 18e), resembling the distinctive fingerprint of AMOC variability[60]. However, the explained variance ratio of AMOC (Supplementary Fig. 18f) has a similar pattern with that from linear trend in the Atlantic Ocean (Supplementary Fig. 9a), indicating that our research period (1970–2018) is still too short to distinguish the contribution of AMOC variability from the linear trend. Recent studies also suggest there may be a significant contribution of anthropogenic climate change in the recent AMO/AMOC changes[16–18]. Consequently, we choose not to include the AMOC/AMO index in MVLR. In fact, we find that including or excluding AMO/AMOC does not change our main results (Supplementary Fig. 18a–d).

**Robustness of regional regressions and choice of time period**. Identifying a statistically significant trend or acceleration strongly depends on the length of study period, as well as starting and ending times. To investigate when the length of study period is reasonably long enough for MVLR model to separate sea-level variability, trend and acceleration, we use large ensemble simulations from the Community Earth System Model (CESM) to identify how robust the climate index regression patterns are to the study period used in the calculation. CESM is a global scale, fully coupled climate model developed by the National Center for Atmospheric Research

(NCAR)[61]. We use historical DSL during 1970–2018 on monthly resolution. For each realization, the natural variability is removed via MVLR (Eqs. 3, 4) with ending year fixed at 2018 and starting years changing from 1970 to 2013. The linear and quadratic terms are estimated at each grid point from each realization, then uncertainty is determined from the STD of all 35 realizations. The results in Supplementary Fig. 3 show the global mean of uncertainties (90% CL) of trend and acceleration. For the acceleration during the historical period, after minimising the natural variability via MVLR, the range of global mean uncertainty increases rapidly as the length of the observations reduces (Supplementary Fig. 3a). For example, the global mean of acceleration uncertainty during 1970–2018 is 0.089 mm yr$^{-2}$, which is much less than 0.548 mm yr$^{-2}$ during satellite era 1993–2018 and 3.960 mm yr$^{-2}$ over the shorter overlapping period 2007–2018 (Supplementary Figs. 3a–d). Similarly, the GMSL trend uncertainty also reduces significantly from 9.9 mm yr$^{-1}$ over short overlapping period 2007–2018 to 1.8 mm yr$^{-1}$ over 1970–2018 (Supplementary Figs. 3e–h). These results agree with previous study, indicating that a data length on the order of 50 years are required for robust regression of regional sea level on low-frequency variability such as PDO[19]. Including the decadal variability in the MVLR model for the altimeter regional sea level is not feasible here due to the current length of the datasets. Therefore, our testing with CESM large ensemble simulations indicates that the 49-year window length (1970–2018) is a reasonable choice for monthly tide-gauge records to estimate linear and quadratic terms via the MVLR model (Eqs. 3, 4) on regional scale. The RCP8.5 scenario over 2007–2032 on annual resolution are used here for future projections. To remove the natural variability, we randomly select 21 CESM realizations (the same model number as the AR5 projection under the RCP8.5 scenario) out of total 35 to estimate the ensemble mean, and repeat this procedure 35 times to derive 35 subsets. Then the trend and acceleration uncertainty can be estimated from the STD of these 35 subsets. The future projection results (Supplementary Fig. 6) are similar to those found for historical period, e.g. the uncertainty of acceleration decays quickly from 0.479 mm yr$^{-2}$ over 2007–2018 to 0.075 mm yr$^{-2}$ with increasing data length ending at 2032. Hence, we conclude that a period of 26-year is feasible for us to detect a robust quadratic term for annual multimodel ensemble-mean outputs from sea-level projections on regional scale.

**Residual VLM correction.** VLM ($V$), the change in the height of the sea floor (in a geocentric reference frame), includes solid Earth deformations in response to last deglaciation (GIA VLM, $V_{GIA}$), ongoing mass redistribution (contemporary GRD VLM, $V_{cGRD}$), as well as other factors like tectonics, sediment compaction and anthropogenic subsidence (residual VLM; $V_r$)[14]. This can be expressed as follows:

$$V = V_{GIA} + V_{cGRD} + V_r \qquad (5)$$

Recognising that satellites measure geocentric sea level whereas tide gauges measure relative sea level, we use the trend of the difference between tide gauge and nearby altimeter measurements at each tide-gauge station to infer local VLM following ref. 7, and the $V$ at each tide-gauge station is estimated by:

$$V = A - R \qquad (6)$$

where $R$ is the RSL time series based on tide-gauge records during 1993–2018, and $A$ is the time series of geocentric sea level from the nearest grid to tide gauges from the satellite altimetry over the same period. We found little difference when comparing our estimated VLM rates with GPS records from the University of La Rochelle solution (ULR6A)[7]. However, the GPS results are noisier and only available at a subset of 103 stations.

The $V_{GIA}$ rates (Fig. 4) used here are from the ICE5G model output, which is the one of the products used for the AR5 projections[40,41,62,63]. We adopt the $V_{cGRD}$ rates and the accompanying uncertainties during 1993–2014 estimated by ref. 46.

To propagate the uncertainties from $V$, $V_{GIA}$, and $V_{cGRD}$ to $V_r$ trend, we assume the uncertainties from each component are independent and are added in quadrature to obtain the final $V_r$ trend uncertainty. The residual VLM ($V_r$) correction is applied to the 177 tide-gauge records (Fig. 4; Supplementary Fig. 10). We assume the rates of residual VLM component are constant over the comparison period, and non-linear VLM (e.g. due to changing rates of extraction of underground water or elastic rebound associated with nearby ice mass loss) is not considered here. The additional uncertainty from this $V_r$ correction is added in quadrature to the uncertainty in the sea-level trend for each tide-gauge station. Then the 90% CL is computed as the one STD uncertainty multiplied by 1.65, assuming a normal distribution.

**Sea-level projections**

*AR5 projection.* We use sea-level projections from the IPCC AR5[1], which give multimodel ensemble-mean sea level, 5% (lower) and 95% (upper) ranges. The AR5 projections are available with annual outputs from 2007 to 2100 on a spatial resolution grid of 1° × 1° grid under the RCPs 2.6, 4.5, and 8.5. The time series under all scenarios are extracted at the nearest grid point to each tide-gauge record.

*SROCC projection.* Compared with AR5 projection, the recently published IPCC SROCC projection includes a new estimate of the dynamic contribution of Antarctic ice-sheet melting[2], the other components are exactly identical to the AR5 projections. The GMSL from SROCC projection under RCP8.5 scenario presents a slightly higher rate of rise than that from AR5 projection during the second half of the 21st century (Supplementary Figs. 5a-c).

*AR5_lp projection.* For the AR5 projections (as well as SROCC projection), the RCP2.6 contains 16 models, and the RCP4.5 and 8.5 are based on 21 models. Since the ensemble size is not large enough for all three RCPs (<30 members) to fully remove the natural variability by averaging across the CMIP5 ensemble[21], we also utilise the AR5_lp projections, which intentionally remove natural variability by applying low-pass filtering to the ocean DSL[22,23]. The AR5_lp projections generally follow the IPCC AR5[64], including the same published GMSL contributions from the IPCC, but with three minor differences. First, the AR5_lp projections use a different GIA-induced RSL change based on a gravitationally self-consistent sea-level theory, which considers time-varying shorelines, changes in the geometry of grounded marine-based ice, and the feedback into sea level of Earth's rotation changes[65,66]. Secondly, more CMIP5 models are included under three scenarios, i.e. RCP2.6 includes 21 models, and RCP4.5 and 8.5 contain 28 models. Lastly but the most importantly, 20-year running-mean low-pass filtering is applied to the historical and projected DSL component with same reference period during 1986–2005. The filtered DSL components is then added to the other sea-level components before deriving the ensemble averaging AR5_lp outputs (more model details are stated in ref. 23).

Through this low-pass filtering, the AR5_lp projections contain less natural variability. To demonstrate this, we used two versions of ensemble-mean annual DSL based on 28 CMIP5 models under RCP4.5, one being low-passed before ensemble averaging and the other not. The RMSD between these two versions (with or without low-pass filtering) during 2007–2100 clearly shows spatial patterns related to internal variability of DSL (Supplementary Fig. 4a). For comparison, the STD of the annual DSL (de-trended over the historical period 1950–2000 for better representation of internal variability) from a single CMIP5 model (CCSM4) indicates typical natural sea-level variability (Supplementary Fig. 4b). Spatial similarities are evident between these two panels (despite lower magnitudes in the upper panel due to ensemble averaging), indicating the natural variability wasn't completely averaged out in the case of no low-pass filtering. This low-pass filtering process is an efficient solution for removing the natural variability, contributing to a better estimation of the anthropogenic signal and the acceleration during a relatively short period. For example, at tide-gauge ID 90 (location shown in Fig. 2), the DSL time series without low-pass filtering contain obvious interannual variability (Supplementary Fig. 19a), which can obscure the estimates of acceleration. The linear-plus-quadratic fits of DSL without low-pass filtering present large diversity (Supplementary Fig. 19b), some models even show deceleration over the short period. After the low-pass filtering, the multimodel ensemble averaging DSL time series contain less natural variability, and the estimation of acceleration has a much smaller ensemble spread (Supplementary Figs. 19c-d).

**Statistical analysis.** We use the non-parametric bootstrapping method to estimate the uncertainty of trend and acceleration. The results are almost unchanged if we use ordinarily least squares (allowing for serial correlation by considering the effective degree of freedom) or the Cochrane–Orcott method[67], but the ordinary least square approach has marginally larger uncertainties compared with that the bootstrapping method.

**Observations.** For the monthly time series from observations (both satellite altimeter and TG records), the serial autocorrelation in sea-level changes strongly bias the confidence interval based on the usual Odinary Least Squares regression[67]. Here we use a method based on bootstrapping in which the generation of surrogate time series with the same serial autocorrelation[68]. The main procedures are:

Step 1: Consider a MVLR model including the ENSO and PDO indexes for example:

$$y_i = \beta_0 + \beta_1 t_i + \beta_2 t_i^2 + \beta_3 \text{ENSO}_i + \beta_4 \text{PDO}_i + \varepsilon_i = \widehat{y}_i + \varepsilon_i \qquad (7)$$

where $y_i$ is sea-level observations, $\beta_*$ are the regression coefficients associated with the linear and quadratic changes with time and each climate index, $\varepsilon_i$ is the residual variability, $\widehat{y}_i$ is the hindcast by the MVLR model, $i = 1, 2, \ldots, N$, and $N$ is the length of the time series.

Step 2: Generate 1000 artificial residual time series ($\varepsilon_i^k$, $k = 1, 2, \ldots, 1000$; $i = 1, 2, \ldots, N$) by using the phase-randomized sampling procedure[68]. This resampling procedure preserves the power spectrum, so the resampled series retains the same autocorrelation function as the original residual series.

Step 3: Add the 1000 artificial residuals ($\varepsilon_i^k$) back to the estimators ($\widehat{y}_i$) to generate 1000 surrogate time series ($y_i^k$):

$$y_i^k = \widehat{y}_i + \varepsilon_i^k, \ (k = 1, 2, \ldots, 1000; \ i = 1, 2, \ldots, N) \qquad (8)$$

Step 4: Linear trend (acceleration) of each surrogate time series ($y_i^k$) is estimated, deriving a pool of 1000 outputted trends (accelerations).

Step 5: The central value of trend (acceleration) is determined from the median of 1000 outputted trends (accelerations), and the associated uncertainty is the STD of 1000 trends (accelerations). The 90% CL is computed as the one STD uncertainty multiplied by 1.65, assuming a normal distribution.

**Sea-level projections**. For AR5, SROCC, and AR5_lp sea-level projections, the sea-level projection outputs provide the multimodel ensemble averaging sea level ($\overline{\eta}_i$) with 5% (lower) and 95% (upper) uncertainty bound ($1.96 \times \delta_i$), where the $\delta_i$ is the STD at each time step and the subscript $i$ refers to the annual time steps. If the annual time series include $M$ annual data points, the bootstrap subsample procedures are as follows[69]:

Step 1: perturbed time series ($p_i$) are generated by randomly sampling from the normal distribution with STD ($\delta_i$) at each time step 1000 times, and the artificial sample time series ($\eta_i$) is given by:

$$\eta_i = \overline{\eta}_i + p_i, \ (i = 1, \ldots, M) \tag{9}$$

Step 2: Linear trend (acceleration) of each sample time series ($\eta_i$) is estimated, deriving a pool of 1000 outputted trends (accelerations).

Step 3: Then the central value of trend (acceleration) is determined from the median of 1000 outputted trends (accelerations), and the associated uncertainty is the STD of 1000 trends (accelerations). The 90% CL is computed as the one STD multiplied by 1.65 assuming a normal distribution.

## Data availability

Monthly global-mean sea-level observation over 1993–2018 from the Australian Commonwealth Scientific and Industrial Research Organization (CSIRO; available online at http://www.cmar.csiro.au/sealevel/N_a_altimetry_gmsl_refined.html)[10] and the U.S. National Aeronautics and Space Administration (NASA) Goddard Space Flight Centre (GSFC; available online at https://podaac.jpl.nasa.gov/dataset/MERGED_TP_J1_OSTM_OST_GMSL_ASCII_V42)[11] are employed. Monthly tide-gauge observations over 1970–2018 are from the Permanent Service for Mean Sea Level (PSMSL; https://www.psmsl.org/) Revised Local Reference (RLR) data[48]. Monthly sea-level reanalysis data during the 1970–2018 with horizontal resolution of 1º are from the European Centre for Medium-Range Weather Forecast (ECMWF) Ocean Reanalysis System 5 (ORAS5; https://www.ecmwf.int/en/research/climate-reanalysis/ocean-reanalysis)[51]. All climate indices used in this study are publicly available and can be downloaded from the corresponding website (the Multivariate El Niño—Southern Oscillation Index[52]; https://www.esrl.noaa.gov/psd/enso/mei/; the Pacific Decadal Oscillation Index[53]; http://research.jisao.washington.edu/pdo/; the Indian Ocean Dipole represented by the Indian Ocean Dipole Mode[54]; https://www.esrl.noaa.gov/psd/gcos_wgsp/Timeseries/DMI/; the Southern Annular Mode Index[55]; http://www.nerc-bas.ac.uk/icd/gjma/sam.html; the North Atlantic Oscillation Index[56]; https://climatedataguide.ucar.edu/climate-data/hurrell-north-atlantic-oscillation-nao-index-pc-based). The Community Earth System Model (CESM)[61] large ensemble simulations developed by the National Center for Atmospheric Research (NCAR) are available at http://www.cesm.ucar.edu/projects/community-projects/LENS/data-sets.html. The Global Positioning System (GPS) records are from the University of La Rochelle solution (ULR6A; https://www.sonel.org/-ULR6a-.html)[7]. The glacial isostatic adjustment (GIA) estimate from the ICE5G model[40] can be found at http://www.atmosp.physics.utoronto.ca/~peltier/data.php. The sea-level projections from Intergovernmental Panel on Climate Change (IPCC) Fifth Assessment Report (AR5; http://icdc.cen.uni-hamburg.de/1/daten/ocean/ar5-slr.html)[1] and Special Report on the Ocean and Cryosphere in a Changing Climate (SROCC; https://www.ipcc.ch/srocc/download-report/)[2] are used over 2007–2032. The regional low-pass filtered AR5 sea-level projections data are available from the corresponding authors on request. GMSL reconstruction datasets are taken from refs. [12],[13],[15]. CW2011 was updated to 2018 by the author with permission to use.

## Code availability

The codes that support the findings of this study are available upon request from the corresponding authors.

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

## Acknowledgements

We acknowledge the IPCC AR5 sea-level projections data distributed by the Integrated Climate Data Centre, and we also thank the CSIRO sea-level group for producing and making available their AR5_lp sea-level projections data. We thank B. Legresy for the bias-corrected satellite altimetry sea-level fields data. We would like to thank C. Watson, D. Monselesan and K. Lyu for detailed comments on an early draft, which helped to improve the manuscript significantly. This work is supported by the Centre for Southern Hemisphere Oceans Research, a joint research centre between the QNLM and the CSIRO, and Australian Research Council's Discovery Project funding scheme (project DP190101173). J.W. is also supported by the China Scholarship Council (201806330014). X.C. is supported by the Natural Science Foundation of China under Grant 41825012 and 41776032. The views expressed herein are those of the authors and are not necessarily those of the Australian Research Council.

## Author contributions

J.A.C. conceived and designed the study. J.W. carried out the analysis and produced all figures under the guidance of J.A.C., X.Z., and X.C. J.W., J.A.C., and X.Z. wrote the first draft of the manuscript, and all authors discussed the results and made substantial contributions to rewriting and revising the manuscript.

## Competing interests

The authors declare no competing interests.
