## [Peer Review File · Nature Communications]

Reviewers' comments:

Reviewer #3 (Remarks to the Author):

Remarks to the Author:

Critical comments to the manuscript entitled "“Observations confirm projected global mean and regional sea-level trends and accelerations” by Wang and co-authors.

The manuscript has been revised, however, there are only some “cosmetic” changes in the text, mainly the way about their results are presented, and there are no considerable changes to address the main issues highlighted in my previous review to the version submitted to the Nature Climate Change journal, e.g.

1. There are numerous speculative statements in the paper, not supported by the results.. or there is misinterpretation of the results (see my comments below)
2. The analysis and results are not particularly rigorous or in-depth (see my comments below);
3. As I have pointed out previously, this manuscript does not provide any considerable step forward in our understanding of sea level changes, and does not bring any new knowledge about future sea level projections. In contrary, authors promote a very confusing message regarding the future trends and acceleration (see my comment below).
4. In this manuscript there is no connection between the results and the statements in abstract, main text and discussion. Title and most of the statements in the main text/abstract are not supported by the outputs from this study.

Regarding 1. There are several speculative statements in the paper, not supported by the results.

Acceleration:

I would like to be very direct - this study does not deliver or demonstrate any statistically significant acceleration in the sea level projections for all three RCPs (please see the table 1), 2007-2032.

a) In addition, the estimates for acceleration in the future sea level projections (e.g. AR5 RCPs, see below some results from table1) are not statistically significant and the conclusion that there is an acceleration during the 2007-2032 is not supported by the results, considering the magnitude of the acceleration, for example 0.035 ± 0.049 ; 0.048 ± 0.047 ; and 0.069 ± 0.048 . Similar case for the SROCC estimates for acceleration.

b) Authors show that future (e.g. period 2007-2032) acceleration is smaller (???) compare to the estimates of acceleration from the observations/reconstructions 1970-2018, calculated as 0.063 ± 0.011 (0.062 ± 0.026), see table 1 (e.g. a part of the table 1 is above). In my opinion (and I suppose for the readers) results suggest that future sea level rise is not accelerating (as the magnitude of acceleration future is smaller compare to the past; + the future acceleration is not statistically significant to make any firm conclusion).

c) Analogous examples could be made for all results of the acceleration in the Table 1.

My concern, that results are suggesting that with increased radiative forcing (e.g. RCPs) and considering already existing acceleration (e.g. estimate of 0.063 +/- 0.011 for the period 1970-2018) a future sea level acceleration (2007-2032) is about 0.035 +/- 0.049; 0.048 +/- 0.047, smaller compare to the already existing (observed) acceleration . It does show that the models are (probably) underestimating future sea level rise... doesn't it? We would expect the magnitude of future sea level acceleration (and trend) to be larger than in the past, considering that radiative forcing is increasing, heat is already in the ocean, glaciers/ice sheets are already losing ice mass ...

Trends:

Again, according to the table 1 (see a part of the table 1 above) the trends for the period 1970-2018 (observations/reconstructions) are the same (the difference between trends is not statistically significant) as for the AR5/SROCC period 2007-2032, all around 3.7 +/- 0.5 and 3.9 +/- 0.6.

There is no statistically significant difference between these trends for the past (1970-2018) and future (2007-2032), that does show (!!!!!) that sea level is not accelerating for the period 2007-2032.. or what is the explanation? If there is no change in the trend (past/future) - sea level is rising with the same rate. However, the past sea level data do show acceleration (0.063 +/- 0.011) and one would expect increase in the trends for future time periods (1) due to already existing acceleration+ response to the forcing (e.g. RCPs) ... There is a confusion (or misleading) in the interpretation of the results, e.g. table 1.

In summary (regarding point 1)

Authors have to provide robust scientific evidence (e.g. numbers, explanations, physical mechanisms) to make any statements.

In this manuscript there is a disconnect between the results and the statements in abstract, main text and discussion. Title and most of the statements in the main text/abstract are not supported by the outputs from this study.

Here is a sentence from the abstract "The confirmation of the projections with observations gives us confidence on our current understanding of near future sea-level change (e.g. sentence from the abstract, and similar statements in the main text).

What kind of understanding is this? What kind of confirmation is that?

Authors show that the trend in future (2007-2032) sea level rise for all scenarios is not statistically different to the trend in the past (1970-2018) (i) ; acceleration for the future (2007-2032) is smaller compare to the past (1970-2018) and in addition, (ii) the estimates of acceleration for the future are not statistically different for all scenarios; and (iii) estimates of acceleration for future sea level projections are not statistically significant.

2. The analysis and results are not particularly rigorous or in-depth (see my comments below);

a) Authors have added a few sentences to their chapter "Method", which are a standard description of the tests.... but it is not a question of the t-test. As I have pointed out above, most of the results

are speculative... see my points about Trends and Acceleration. The quality of statistical analysis in this paper is exceptionally poor, the simple questions about the statistical significance and testing the difference between the trends, e.g. is the difference between these trends in table 1 (e.g. 3.7 ± 0.3 and 3.8 ± 0.5 and so on) are statistically significant are not presented and explained in the text and in the methods.

Estimating the magnitude of acceleration for such show time period (2007-2032) and getting such unclear/confusing results (e.g. none of accelerations are statistically significant) leads to question the value of this manuscript.

b) !!! In addition, authors do not reply to the critical comments from my previous review regarding the selection of time periods, e.g. they use of the different length of time series, their selection of the projections (e.g. from the figure it is clear that projections by Kopp et al, 2014 will are very linear and would not show any acceleration for the early 21st century), reconstructions (e.g. why only two reconstructions are used here???), why these reconstructions are limited to 1970-2018, but longer than 2007-2032?

c) As I have mentioned in my previous review, I question the robustness of the results (e.g. acceleration for individual tide gauge locations) in this study due to lack of clarity about their approach to deal with uncertainties, e.g. numerous corrections were applied to the observations. In addition, observational data were filtered/smoothed... however, no information were provided regarding the propagation of errors due to multiple corrections, and in some case errors were clearly not estimates, please see my next comment (or e.g. please read the second part of abstract, in which trend/accelerations are given without any errors/uncertainties).

3. As I have pointed our previously, this manuscript does not provide any considerable step forward in our understanding of sea level changes. In contrary, author promote a very confusing message regarding the trends and acceleration (see my comments about), a very confusing statements regarding our approach to simulate our future projections, considering that results in this manuscript suggest no changes in sea level trend for the future decades (2007-2032) and no statistically significant acceleration for all RCPs .

4. I am disappointed that authors have not not clarified the most of my comments, providing some 'cosmetic changes" into the manuscript only. The quality of the results, presentation and quality of the figures are poor.

Reviewer #4 (Remarks to the Author):

This study, which objective is to compare linear trend and aceleration of sea level projections with observations at global and regional scales, is a resubmission of a previous manuscript. I read the evaluations of the 3 reviewers concerning the original manuscript as well as the detailed responses of the authors, and find that the latter have accounted for most of the reviewers' comments

(although I have not seen the original manuscript). However from my reading of the current version, I share the basic comment made by the 3 previous reviewers: the periods considered for comparing the data and the projections are significantly different. The whole study is built on the assumption that the trend and acceleration of the projections for 2007-2032 should be the same as in the observations during 1993-2018. If the authors find good agreement (as it is the case), then they conclude for the robustness of the projections. However, there is no guarantee that in the real world, some runaway land ice melt from the ice sheets (e.g., west Antarctica) will not occur (even if not very likely in the next few decades). If this happens, it will give rise to an increased acceleration in the global mean sea level. Thus current agreement between near future projections and past observations would just mean that the latter are too conservative. To me, a more convincing approach would have been to compare projections and observations over the same time span (e.g. the altimetry era) by building a projection record based on the combination of historical runs for 1993-2007 and pure projection beyond 2007. I am not an expert of coupled climate simulations and this approach may have some drawbacks. But the authors should at least mention this possibility and argue why that decided to not follow it.

That being said, I recognize that this study represents a lot of new work and merits to be published, even if I think the conclusions are disputable.

I have a few minor comments:

- Line 102: for the non experts, explain that the difference between DSL and total sea level
- line 146: why 2 acceleration values for CSIRO data?
- Lines 152-156: how do you explain that the corrected GMSL acceleration is twice lower than other published values (e.g., Nerem et al., 2018 who also corrected for natural variability and Pinatubo)?
- Line 186: I suspect that Fig.3 is related to ORA S5. Add this info line 186 as well as the analysis time span.
- Lines 204-208: I suggest you provide deeper discussion about the causes of non uniform acceleration (Fig.3g). The explanation provided here relates the Southern Ocean acceleration to recent warming. Is there a consensus on that? What about other regions like North Pacific, Western Indian Ocean?
- Lines 212-229: Why not correct for GPS-based VLM where available?
- Line 241 (and methods): Did you check whether nearby tide gauge show the same dependency on climate modes?
- Lines 284-289: Here again, some discussion about the causes of regional variability in acceleration would be useful.
- Discussion section: As I mentioned above, the fact that trend and acceleration in projections (up to 2032) agree with past observations just mean to me that projections are rather conservative in terms of future contribution from the ice sheets. This is certainly the best that can be done today but this does not preclude from future surprises in the real world. Such comparisons should be regularly reproduced.
- Table 1: The date for 'Altimeter GMSL' is wrong (not 1993-2018 but 2007-2018; over 1993-2018, the altimeter rate is only 3.2 mm/yr).

If you wish to transfer your manuscript to Scientific Reports, please use our <https://mts-comms.nature.com/cgi->

bin/main.plex?el=A6S7BmZP5A1BTrK2X6A9ftdPUgqJNsrcmmQyuq5pnkDQZ>manuscript transfer portal to initiate the transfer to this journal (or to another journal of your choice in the Nature Research portfolio). If you transfer to Nature-branded journals or to the Communications journals, you will not have to re-supply manuscript metadata and files. This link can only be used once and remains active until used.

All Nature Research journals are editorially independent, and the decision to consider your manuscript will be taken by their own editorial staff. For more information, please see our manuscript transfer FAQ page.

Note that any decision to opt in to In Review at the original journal is not sent to the receiving journal on transfer. You can opt in to In Review at receiving journals that support this service by choosing to modify your manuscript on transfer. In Review is available for primary research manuscript types only.

Reply to the reviewers

In the response to reviewers, the original comments are in black italics (copied verbatim from the reviewer's comments) and our responses are in blue and indented. Unless otherwise noted, line numbers are in the revised (clean/non-tracked changes) manuscript.

Response to reviewer #3:

Major comments:

Remarks to the Author:

Critical comments to the manuscript entitled "Observations confirm projected global mean and regional sea-level trends and accelerations" by Wang and co-authors.

*The manuscript has been revised, however, there are **only** some "cosmetic" changes in the text, mainly the way about their results are presented, and there are no considerable changes to address the main issues highlighted in my previous review to the version submitted to the Nature Climate Change journal, e.g.*

- 1. There are several speculative statements in the paper, not supported by the results.. or there is misinterpretation of the results (see my comments below)*
- 2. The analysis and results are not particularly rigorous or in-depth (see my comments below);*
- 3. As I have pointed out previously, this manuscript does not provide any considerable step forward in our understanding of sea level changes, and does not bring any new knowledge about future sea level projections. In contrary, authors promote a very confusing message regarding the future trends and acceleration (see my comment below).*
- 4. In this manuscript there is no connection between the results and the statements in abstract, main text and discussion. Title and most of the statements in the main text/abstract are not supported by the outputs from this study.*

We believe that Reviewer 3 misunderstood our results and the context. Here, we respond to further explain and clear up any misunderstandings. Firstly, we address the questions of (i) whether our manuscript offers anything new, and (ii) values of the accelerations in the projections and observations.

(i) The value of the manuscript:

Past papers (including several that JC has been a co-author on) have attempted to evaluate historical model sea level simulations, as referred to in the manuscript. These publications have either focussed on global mean sea level or sea level at a small number of tide gauges.

In particular, there have been two papers that have attempted to evaluate the projections of the IPCC Third Assessment Report (Rahmstorf et al. 2007) and the IPCC Fourth Assessment Report (Church et al. 2011). Both of these papers (JC was a co-author/lead author on both) only attempted to evaluate global mean sea level projections and not coastal sea level projections which are important for understanding the impacts of sea level change.

There has been one paper that has attempted to evaluate the IPCC Fifth Assessment Report sea level projections. Watson (2018) argued that the IPCC AR5 projections exceeded the observed rate of sea level rise at 19 tide gauge stations by 1.6 to 2.5 mm/yr. This is not consistent with the community's previous attempts to model 20th century sea levels. We could have made this the focus of our paper but instead chose to highlight the more important issue of evaluation of the AR5 projections and only referred to Watson near the end of our manuscript.

As a convening lead author on previous IPCC sea level chapters, JC would very much welcome any attempts to evaluate previous IPCC projections. Indeed, the lead authors on the upcoming IPCC 6th Assessment Report would very much like to refer to our work in their chapter and to include a figure and the Lead Authors have enquired several times about the status of this manuscript.

We also note that referees 1, 2 and 4 all note the value of our work, although raising a number of issues that we have attempted to deal with. Their comments include:

Reviewer #1: ‘As a general comment, I understand the motivation for this study and do feel like it is a worthwhile exercise. Both (1) evaluating the agreement of observations with available projections and (2) working towards an assessment of which pathway we might be on are of high importance and significance.’

Reviewer #2: ‘I mostly like this paper, but SROCC just came out with new projections.’

Reviewer #4: ‘This is certainly the best that can be done today but this does not preclude from future surprises in the real world. Such comparisons should be regularly reproduced.’

Reviewers 1, 2 and 3 all note that there were more recent projections than the IPCC AR5 projections and the editor requested inclusion of the IPCC SROCC projections in a revised manuscript.

Accordingly, we extended our comparison to include the most recent IPCC SROCC projections in the revised Nature Communications’ manuscript. As stated in our revision, the SROCC projections are slightly updated from the AR5 projections, since only Antarctica contribution were updated and all other contributions remain unchanged.

In agreement with three of the four reviewers and the current IPCC authors, we believe that our manuscript fills an important niche complementing previous publications and it is important to complete a systematic study comparing sea level observations and AR5/SROCC projections both regionally and globally before the IPCC AR6 projections are published, especially in light of the Watson (2018) publication. Our manuscript is not only important scientifically, but is also important in demonstrating the value and weaknesses of the IPCC Projections.

(ii) values of the trends and accelerations in the projections and observations

Based on Reviewer 3’s detailed comments, we think Reviewer 3 misunderstands our results, and we address these issues point-by-point below.

For example, the trends in Table 1 compare historical observations and projections over the common period (2007-2018), and not over 1970-2018 for observation and 2007-2032 for projection. These longer periods were used to help reduce the natural variability that confounds trends over short periods. To avoid further confusion, we have deleted the lines that indicated the longer period over which the multivariable linear regression was applied from the table. Table 1 is a direct support for our main conclusion and clearly gives the values of trends and corresponding uncertainties, as noted in the detailed response below.

The question of the accelerations is addressed in more detail below and we revise the emphasis of the manuscript in light of the reviewer’s comments.

Reviewer 3’s points 1 and 2 are addressed in responses to detailed comments below. With regard to point 4, all the results in the abstract are contained in the text of the manuscript.

We agree with reviewer 3 that the accelerations are for different periods with a period of overlap. As a result, we have changed the text so that we are not stating the observed and projected accelerations are the same but rather that the observed accelerations are larger than those that would be consistent with the Paris goals that require emissions to follow the RCP2.6 scenario (or lower).

Detailed comment 1:

Regarding 1. There are several speculative statements in the paper, not supported by the results.

Acceleration:

I would like to be very direct - this study does not deliver or demonstrate any statistically significant acceleration in the sea level projections for all three RCPs (please see the table 1), 2007-2032.

a) In addition, the estimates for acceleration in the future sea level projections (e.g. AR5 RCPs, see below some results from table1) are **not statistically significant** and the conclusion that there is an acceleration during the 2007-2032 is not supported by the results, considering the magnitude of the acceleration, for example 0.035 ± 0.049 ; 0.048 ± 0.047 ; and 0.069 ± 0.048 . Similar case for the SROCC estimates for acceleration.

GMSL reconstruction (1970-2018)		
D2019	3.7 ± 0.3	0.063 ± 0.011
CW2011	3.7 ± 0.5	0.062 ± 0.026
AR5 projections GMSL (2007-2032)		
AR5 RCP2.6	3.9 ± 0.6	0.035 ± 0.049
AR5 RCP4.5	3.8 ± 0.5	0.048 ± 0.047
AR5 RCP8.5	3.9 ± 0.6	0.069 ± 0.048

Reviewer 3 must mean that the accelerations of the projections are not significantly different from zero. But that is not the most important test. The focus of the manuscript is whether or not the projected trends and accelerations are significantly different to the observed trends and accelerations. Also, please note that we gave 90% confidence limits on the accelerations. So based on the original data from IPCC AR5/SROCC, the accelerations of Global Mean Sea Level (GMSL) over 2007-2032 under RCPs 2.6 and 4.5 are not statistically different from zero with the 90% confidence limits, but RCP 8.5 acceleration is statistically different from zero (less than 5% chance of zero or smaller acceleration). That is, the confidence limits mean that there is less than 17% chance of zero or negative accelerations for RCP4.5 and RCP2.6 (the central values are greater than one standard deviation from zero) and for RCP 8.5 acceleration there is less than 5% chance of zero or smaller acceleration. These acceleration uncertainties are determined by the short period necessary for comparison and the uncertainties decrease with longer periods (as shown in supplementary Figs. 3 and 6). As indicated in Figure 1b below, the differences in accelerations between the scenarios are systematic and continue into the future. This figure has been added to the Supplementary Figures.

We are careful not to overstate our results by reporting the computed acceleration uncertainty range, e.g., ‘... However, because of the short period available for comparison, the regional accelerations are not yet statistically different to the projections for any of the scenarios’ (lines 345-347 in discussion section). Furthermore, we always compare accelerations (or trends) with uncertainty range throughout our paper.

Authors show that future (e.g. period 2007-2032) acceleration is **smaller** (????!!) compare to the estimates of acceleration from the observations/reconstructions 1970- 2018, calculated as 0.063 ± 0.011 (0.062 ± 0.026), see table 1 (e.g. a part of the table 1 is above). In my opinion (and I suppose for the readers) results suggest that future sea level rise is not accelerating (as the magnitude of acceleration future is **smaller** compare to the past; + the future acceleration is not statistically significant to make any firm conclusion).

b) Analogous examples could be made for all results of the acceleration in the Table 1.

My concern, that results are suggesting that with increased radiative forcing (e.g. RCPs) and

considering already existing acceleration (e.g. estimate of 0.063 ± 0.011 for the period 1970-2018) a future sea level acceleration (2007-2032) is about 0.035 ± 0.049 ; 0.048 ± 0.047 , smaller compare to the already existing (observed) acceleration . It does show that the models are (probably) underestimating future sea level rise... doesn't it? We would expect the magnitude of future sea level acceleration (and trend) to be larger than in the past, considering that radiative forcing if increasing, heat is already in the ocean, glaciers/ice sheets are already losing ice mass ...

Firstly, why would we expect apriori the magnitude of future sea level acceleration be larger than in the past? Future sea level scenarios, including any acceleration, must be considered in conjunction with the emission scenarios and different study periods. Emission mitigation scenario RCP 2.6 requires that carbon dioxide (CO₂) emissions start to decline by 2020 and go to zero prior to 2100. For GMSL under RCP 2.6, the acceleration is declining in future and even becomes negative after 2052 (Figure 1b below). This is the whole point of mitigation scenarios – they are an attempt to slow (decelerate) the rate of climate change and sea level rise. It is incorrect to assume that future accelerations must be larger than in the past.

Figure 1 (a). Time series of GMSL from observations and IPCC AR5 projections under RCP 2.6, RCP 4.5 and RCP 8.5 scenarios. The shaded region is the likely range of projection. (b). Boxes denotes acceleration derived over 1993-2018 from satellite observation and over 1970-2018 from reconstructions of GMSL with 90% confidence level, with natural variability removed using a multiple variable linear regression model. Lines starting from 2020 indicate projected acceleration derived from 25-year moving quadratic fits under three RCP scenarios. Error bars denote the projected acceleration uncertainties during our research period (2007-2032) with 90% confidence level (including uncertainty from the multiple-model spread shown in Figure a). Note, this figure has been added to the Supplementary Figures.

Secondly, we compare historical sea level accelerations with sea level projections under three RCP scenarios in our paper, including RCPs 2.6, 4.5 and 8.5. The central values of observed acceleration (1970 to 2015) based on D2019 reconstruction (0.063 ± 0.011 mm yr⁻²) is larger than the projected

(2007-2032) acceleration under RCP 2.6 ($0.035 \pm 0.049 \text{ mm yr}^{-2}$) and RCP 4.5 ($0.048 \pm 0.047 \text{ mm yr}^{-2}$) but it is smaller than the acceleration under RCP 8.5 (0.069 ± 0.048).

Also,

'It does show that the models are (probably) underestimating future sea level rise...doesn't it? We would expect the magnitude of future sea level acceleration (and trend) to be larger than in the past, considering that radiative forcing is increasing, heat is already in the ocean, glaciers/ice sheets are already losing ice mass'

Again, for RCP2.6, we should NOT expect the future acceleration to be larger than in the past (Figure 1b) and the statements that “*the models are underestimating future sea level rise*” is not supported by any justification. Mitigation emission scenarios are designed to slow the rate of climate change.

Trends:

*Again, according to the table 1 (see a part of the table 1 above) the trends for the period 1970- 2018 (observations/reconstructions) are **the same (the difference between trends is not statistically significant)** as for the AR5/SROCC period 2007-2032, all around 3.7 ± 0.5 and 3.9 ± 0.6 .*

*There is no statistically significant difference between these trends for the past (1970-2018) and future (2007-2032), that does show (!!!!) that **sea level is not accelerating** for the period 2007-2032.. or what is the explanation? If there is no change in the trend (past/future)*

- sea level is rising with the same rate. However, the past sea level data do show acceleration (0.063 ± 0.011) and one would expect increase in the trends for future time periods (1) due to already existing acceleration+ response to the forcing (e.g. RCPs) ... There is a confusion (or misleading) in the interpretation of the results, e.g. table 1.

As stated in Table 1 caption: ‘Global mean and regional weighted mean sea-level trends (mm yr^{-1}) over the common period (2007-2018)’ (lines 842-843), here we compare trends over the **common period (2007-2018)** of observation and projections, not 1970-2018 for observation and 2007-2032 for AR5/SROCC projection. The longer periods (1970-2018 and 2007-2032) were only used to minimise the climate variability noise in the observations and to estimate an observed acceleration. Based on the results that there is no statistically significant difference between trends from observation and future projection over the common period 2007-2018, we draw the conclusion: “After considering the impacts of natural variability and local vertical land motion (VLM), the observed trends from both GMSL and regional weighted mean at 177 tide-gauge stations distributed globally confirm the projections under three Representative Concentration Pathway (RCP) scenarios are within the 90% confidence level” (lines 23-27 in abstract). Reviewer 3’s has misquoted Table 1. And yes, all three RCP trends are essentially the same over 2007-2032, as we state, but that does not imply there are not accelerations in the observations and projections.

In summary (regarding point 1)

Authors have to provide robust scientific evidence (e.g. numbers, explanations, physical mechanisms) to make any statements.

In this manuscript there is a disconnect between the results and the statements in abstract, main text and discussion. Title and most of the statements in the main text/abstract are not supported by the outputs from this study.

*Here is a sentence from the abstract “The confirmation of the projections with observations gives us confidence on **our current understanding of near future sea-level change** (e.g.*

sentence from the abstract, and similar statements in the main text).

What kind of understanding is this? What kind of confirmation is that?

Authors show that the trend in future (2007-2032) sea level rise for all scenarios **is not statistically different to the trend in the past** (1970-2018)

(i) ; acceleration for the future (2007-2032) **is smaller** compare to the past (1970-2018) and in addition, (ii) the estimates of acceleration for the future **are not statistically different for all scenarios**; and (iii) estimates of acceleration for future sea level projections **are not statistically significant**.

We have removed some subheadings from Table 1 which may have misled reviewer 3 as to what periods the trends are computed for.

We do not show results that the “trend in future (2007-2032) sea level rise for all scenarios is not statistically different to the trend in the past (1970-2018)”. Rather, our results compare trends over the common period (2007-2018) of observation and projection. Since there is no statistically significant difference between trends from observation and future projection over 2007-2018 (Table 1, second column), we draw the conclusion: “..., the observed trends from both GMSL and regional weighted mean at 177 tide-gauge stations distributed globally confirm the projections under three Representative Concentration Pathway (RCP) scenarios within 90% confidence level” (lines 23-27 in abstract). Again, Reviewer 3’s misquoted Table 1.

(i) The central value of the acceleration under RCP8.5 (0.069 ± 0.048) in the future (2007-2032) is larger compare to the past (0.063 ± 0.011 of D2019 reconstruction over 1970-2018). Please see reply under Detailed comment 1 (b).

(ii) During our research period (2007-2032), the trends for the three RCP scenarios have not started to diverge significantly (Figure 1a) even though there are different accelerations.

(iii) Please see reply under Detailed comment 1 (a).

Detailed comments 2:

2. *The analysis and results are not particularly rigorous or in-depth (see my comments below);*

a) *Authors have added a few sentences to their chapter “Method”, which are a standard description of the tests.... but it is not a question of the t-test. As I have pointed out above, most of the results are speculative... see my points about Trends and Acceleration. The quality of statistical analysis in this paper is exceptionally poor, the simple questions about the statistical significance and testing the difference between the trends, e.g. is the difference between these trends in table 1 (e.g. 3.7 ± 0.3 and 3.8 ± 0.5 and so on) are statistically significant are not presented and explained in the text and in the methods.*

Correct, the trends are not statistically significant, as we stated. Please see detailed reply about **Trends and Acceleration** under Detailed comment 1.

Also,

“the simple questions about the statistical significance and testing the difference between the trends, e.g. is the difference between these trends in table 1 (e.g. 3.7 ± 0.3 and 3.8 ± 0.5 and so on) are statistically significant are not presented and explained in the text and in the methods”

This comment is also based on misquotation of our results in Table 1. In fact, we compare trends

over the common period (2007-2018) of observation and projection, and the conclusion based on trends in Table 1 is that there is no statistical difference between observed and projected trends over the common period (2007-2018): ‘After considering the impacts of natural variability and local vertical land motion (VLM), the observed trends from both GMSL and regional weighted mean at 177 tide-gauge stations distributed globally confirm the projections under three Representative Concentration Pathway (RCP) scenarios within 90% confidence level’ (Lines 23-27).

Estimating the magnitude of acceleration for such show time period (2007-2032) and getting such unclear/confusing results (e.g. none of accelerations are statistically significant) leads to question the value of this manuscript.

b) *!!! In addition, authors do not reply to the critical comments from my previous review regarding the selection of time periods, e.g. they use of the different length of time series, their selection of the projections (e.g. from the figure it is clear that projections by Kopp et al, 2014 will are very linear and would not show any acceleration for the early 21st century), reconstructions (e.g. why only two reconstructions are used here???), why these reconstructions are limited to 1970-2018, but longer than 2007- 2032?*

We replied about the selection of different time period, which is clearly present in our previous ‘Reply_to_Reviewers_NCC.pdf’ file in major revisions section point (2). We repeat a similar explanation here:

Because of the complication of natural variability, it is not possible to separate the acceleration from the decadal (multi-decadal) variability during this short 12 years. Therefore, we extend the data period for both observations and sea-level projections so that they have sufficiently long data length for a robust determination of the acceleration with sufficiently small uncertainty and natural variability removed. In the revision, to fully quantify this, we added more testing with the CESM large ensemble simulation, which was summarized in a new subsection in Methods named ‘Robustness of regional regressions and choice of time period’ and Supplementary Figs. 3, 6. Through this new subsection, we give the reason for our choices on the different comparison periods and address the robustness our estimated trends and accelerations.

It is not reasonable to state that there are no accelerations in the projections for the early 21st century. As indicated in Figure 1 above, accelerations either decrease or increase relatively slowly and smoothly for RCP2.6 and RCP8.5, respectively, before 2050. As a result, there is a growing difference in the accelerations between the RCP scenarios and the observed acceleration is consistent with the projected future RCP4.5 – RCP8.5 accelerations but will need to be reduced to be consistent with the lower and falling RCP2.6 acceleration (which is closer to the Paris targets).

We agree with the reviewers that the accelerations are for different periods with a period of overlap. As a result, we have changed the text so that we are not stating the observed and projected accelerations are the same but rather that the observed accelerations are larger than those that would be consistent with the Paris goals that require emissions to follow the RCP2.6 scenario (or lower).

(e.g. why only two reconstructions are used here???)

The period since 1970 is chosen as this is the time when anthropogenic forcing is responsible for the majority of the sea level change (see line 53; Slangen et al., 2017) and it gives a long enough period to substantially reduce the uncertainty from natural variability. We did not use additional reconstructions because there were only small differences among different reconstructions since 1970 (lines 74-76).

(e.g. from the figure it is clear that projections by Kopp et al, 2014 will are very linear and would not show any acceleration for the early 21st century)

As shown in Figure 2 (Fig. A2 in ‘Reply_to_Reviewers_NCC.pdf’ file), the Kopp projection (2014, 2017) is available on decadal temporal resolution and contains only three temporal steps during our research period (2007-2032; Figure 2d-f). It is not reasonable to conclude that an only 3-point lines ‘are very linear and would not show any acceleration’. Also, this is inconsistent with the IPCC projections which do have an acceleration, as can be most clearly seen for RCP8.5.

Figure 2. GMSL time series from IPCC AR5 and Kopp projections relative to 1986–2005. The lines show the median projections under RCPs 2.6, 4.5 and 8.5 based on annual AR5 (red) and decadal Kopp (blue) over 2007-2100 (a-c) and 2007-2032 (d-f) respectively. The likely range is shown as a shaded band.

c) *As I have mentioned in my previous review, I question the robustness of the results (e.g. acceleration for individual tide gauge locations) in this study due to lack of clarity about their approach to deal with uncertainties, e.g. numerous corrections were applied to the observations. In addition, observational data were filtered/smoothed... however, no information were provided regarding the propagation of errors due to multiple corrections, and in some case errors were clearly not estimates, please see my next comment (or e.g. please read the second part of abstract, in which trend/accelerations are given without any errors/uncertainties).*

The only corrections applied to the tide gauge data were:

- Locations with major data gaps were removed from our data set.
- For all monthly data used in this study, the long-term mean seasonal cycle over the whole period is removed and a 5-month running-mean filter is applied before further analysis.
- Vertical land motion corrections were applied. These are constant through the record thus have no impact on the computed accelerations.

We have rewritten the *Statistical analysis* in Methods to clarify the calculation process of uncertainty, and the reply to Reviewer 3’s comment is clearly present in our previous ‘Reply_to_Reviewers_NCC.pdf’ file in major revisions section point (3). Here we point out a more detailed explanation in our manuscripts:

- 1) Uncertainty from VLM-related correction (including GIA and contemporary GRD corrections): For observed GMSL, ‘We use error estimates of 0.06 mm yr^{-1} for the GIA correction from ref. ⁴⁷ and 0.01 mm yr^{-1} for the contemporary GRD correction from ref. ⁴⁶ (one standard deviation) and add these in quadrature to the uncertainties of GMSL trend. Then the 90% CL is computed as the one standard deviation uncertainties multiplied by 1.65, assuming a normal distribution’ (lines 508-512).

For tide-gauge observation, ‘To propagate the uncertainties from V , V_{GLA} and V_{cGRD} to V_r trend, we assume the uncertainties from each component are independent and are added in quadrature to obtain the final V_r trend uncertainty. The residual VLM (V_r) correction is applied to the 177 tide-gauge records (Fig. 4; Supplementary Fig. 9). We assume the rates of residual VLM components are constant over the comparison period, and non-linear VLM (e.g., due to changing rates of extraction of underground water or elastic rebound associated with nearby ice mass loss) is not considered here. The additional uncertainty from this V_r correction is added in quadrature to the uncertainty in the sea-level trend for each tide-gauge station. Then the 90% CL is computed as the one standard deviation uncertainties multiplied by 1.65, assuming a normal distribution’ (lines 682-690).

2) Uncertainty from MVLR model:

For observations, ‘The statistical uncertainties of each regression coefficient in the least squares regression are estimated following ref.⁵⁷ using a two-sided student’s t-test, with auto-correlation of the sea-level time series (both observations and projections) taken into consideration. Thus, the number of independent samples (effective degree of freedom, N_e) is smaller than the total number of samples (N). ... The 90% CL is computed as the one standard deviation uncertainties multiplied by 1.65, assuming a normal distribution’ (lines 749-759).

For projections, ‘For AR5, SROCC and AR5_lp sea-level projections, the uncertainties in the regression coefficients come from not only the least square calculation as described above but also from the multiple-model spread. Here the bootstrap method is used to estimate the statistical confidence level⁶⁶. ... The final uncertainty of sea-level projections is derived by adding uncertainty of multiple-model spread and uncertainty of the least square calculation in quadrature. The 90% CL is computed as the final uncertainty multiplied by 1.65 assuming a normal distribution’ (lines 761-778).

Also, the abstract has already been revised to include the uncertainties (Line 27; Lines 29-32).

Detailed comments 3:

3. *As I have pointed out previously, this manuscript does not provide any considerable step forward in our understanding of sea level changes. In contrary, author promote a very confusing message regarding the trends and acceleration (see my comments about), a very confusing statements regarding our approach to simulate our future projections, considering that results in this manuscript suggest no changes in sea level trend for the future decades (2007-2032) and no statistically significant acceleration for all RCPs.*

For the value of the manuscript, please see our response to Major comments above.

considering that results in this manuscript suggest no changes in sea level trend for the future decades (2007-2032) and no statistically significant acceleration for all RCPs.

We do not show results for “*sea level trend for the future decades (2007-2032)*”, our results compare projected trends over common period (2007-2018) with observations over the same period (Table 1 caption, lines 842-843), thus this comment is not valid. The acceleration uncertainties are determined by the short period available for comparison, please see detailed reply under Detailed comment 1 a.

Detailed comments 4:

4. *I am disappointed that authors have not clarified the most of my comments, providing some ‘cosmetic changes’ into the manuscript only. The quality of the results, presentation and quality of the figures are poor.*

We responded to earlier comments point by point in ‘Reply_to_Reviewers_NCC.pdf’ file, and here we have given some more detailed explanations.

The resolution of all figure in manuscript is 600 dpi (Nature Communication format: 300 dpi or higher resolution).

Response to reviewer #4:

Reviewer #4 (Remarks to the Author):

This study, which objective is to compare linear trend and acceleration of sea level projections with observations at global and regional scales, is a resubmission of a previous manuscript. I read the evaluations of the 3 reviewers concerning the original manuscript as well as the detailed responses of the authors and find that the latter have accounted for most of the reviewers' comments (although I have not seen the original manuscript).

Thank you for your positive comment on our previous effort in addressing reviewers' comments!

However, from my reading of the current version, I share the basic comment made by the 3 previous reviewers: the periods considered for comparing the data and the projections are significantly different. The whole study is built on the assumption that the trend and acceleration of the projections for 2007-2032 should be the same as in the observations during 1993-2018. If the authors find good agreement (as it is the case), then they conclude for the robustness of the projections.

Ideally, we should compare sea-level trends and projections and observations within the overlapping period (2007-2018). We do this for the trends. However, because of the complication of natural variability, it is very challenging to separate the acceleration from the decadal (multi-decadal) variability during this short 12 years. Therefore, we extend the data period for both observations and sea-level projections so that they have sufficiently long data length for a robust determination of the acceleration with natural variability removed. In the revision, to fully quantify this, we added more testing with the CESM large ensemble simulation, which was summarized in subsection in Methods named 'Robustness of regional regressions and choice of time period' and Supplementary Figs. 3, 6. In this subsection, we give the reason for our choices on the different comparison periods and address the robustness our estimated trends and accelerations.

We have also changed the text so that we are not stating the observed and projected accelerations are the same but rather that the observed accelerations are larger than those that would be consistent with the Paris goals that require emissions to follow the RCP2.6 scenario (or lower).

However, there is no guarantee that in the real world, some runaway land ice melt from the ice sheets (e.g., west Antarctica) will not occur (even if not very likely in the next few decades). If this happens, it will give rise to an increased acceleration in the global mean sea level. Thus, current agreement between near future projections and past observations would just mean that the latter are too conservative.

We concur that the agreement between the observations and the projections does not guarantee there will be no rapid acceleration in ice sheet contributions later in the 20th century, and indeed we referred to this possibility (lines 36-37 and lines 358-361) of the previous manuscript, and have slightly strengthened the conclusion in this revised version (lines 32-37 in revision). However, recent publications (Edwards et al. 2019; Golledge et al. 2019) of ice sheet contributions included in the IPCC SROCC projections result in only modestly higher projections than the AR5 assessment.

We have reworded the text regarding the acceleration. Rather than trying to argue that the observed and projected accelerations were equivalent, we now state that the observed historical acceleration is consistent with the projected future RCP4.5 – RCP8.5 accelerations but that the

real sea level acceleration will need to be reduced to be consistent with the lower and falling RCP2.6 acceleration and the Paris targets. Accordingly, we have softened the title to “Reconciling global mean and regional sea level change in projections and observations”.

To me, a more convincing approach would have been to compare projections and observations over the same time span (e.g. the altimetry era) by building a projection record based on the combination of historical runs for 1993-2007 and pure projection beyond 2007. I am not an expert of coupled climate simulations and this approach may have some drawbacks. But the authors should at least mention this possibility and argue why that decided to not follow it..

As discussed in the second paragraph of the manuscript, past papers (including several that JC has been a co-author on) have attempted to evaluate historical model sea level simulations as a way to evaluate our understanding. This is similar to, but not identical with, the Reviewer’s suggestion.

However, here the attempt was to do something different. Specifically, we wished to evaluate the *projections* as published by IPCC. We wanted this evaluation to be as completely independent of the historical simulations that are affected by observations, at least in the specification of radiative forcing.

Historically, there have been two papers that have attempted to evaluate the projections of the IPCC Third Assessment Report (Rahmstorf et al. 2007) and the IPCC Fourth Assessment Report (Church et al. 2011). Both of these papers (JC was a co-author/lead author on both) only attempted to evaluate global mean sea level projections and not coastal sea level projections which are important for understanding the impacts of sea level change.

There has been one paper that has attempted to evaluate the IPCC Fifth Assessment Report sea level projections. Watson (2018) argued that the IPCC AR5 *projections* exceeded the observed rate of sea level rise at 19 tide gauge stations by 1.6 to 2.5 mm/yr. This is not consistent with the community’s previous attempts to model 20th century sea levels. We could have made this the focus of our paper but instead chose to highlight the more important issue of evaluation of the AR5 projections rather than focus on a criticism of Watson, and only referred to Watson near the end of our manuscript.

As a convening lead author on previous IPCC sea level chapters, JC would very much welcome any attempts to evaluate previous IPCC projections. Indeed, the lead authors on the upcoming IPCC 6th Assessment Report would very much like to refer to our work in their chapter and to include a figure and the Lead Authors enquired about the status of this manuscript.

Secondly, the proposed approach suggested by the reviewer is hard to achieve due to the data availability at present. The AR5 projection provides global mean and regional sea level including several components (such as ocean thermal expansion, mass loss from glaciers and ice sheets, terrestrial water storage). Several components are simulated separately and summed up together to derive the total sea level. Because of this, some components are only available in the future projection period. For example, the ice sheet component is simulated relative to the reference period 1986-2005 and starts from zero. Although we could manage to get all components based on other dataset during historical period, it’s hard to merge the historical datasets (either based on observation or model simulation) with future projection at 2007 due to the complication of natural variability (especially on regional scale) and this would no longer be an evaluation of the projections alone.

That being said, I recognize that this study represents a lot of new work and merits to be published, even if I think the conclusions are disputable.

Thank you.

I have a few minor comments:

- Line 102: for the non-experts, explain that the difference between DSL and total sea level

Done. We add explanation as: “The DSL is the sea level respect to the geoid that is determined by the dynamical process associated with ocean density and circulation (Gregory et al., 2019), while total sea level also include the changes in the mass of ocean associated with glacier and ice sheet mass loss or terrestrial water storage changes” (lines 109-112 in revision).

-line 146: why 2 acceleration values for CSIRO data?

The accelerations are from CSIRO GIA-adjusted and GPS-adjusted data respectively. Line 153-154 are revised to clarify this in the revision.

-Lines 152-156: how do you explain that the corrected GMSL acceleration is twice lower than other published values (e.g., Nerem et al., 2018 who also corrected for natural variability and Pinatubo)?

As discussed in lines 152-158, after removing the natural variability only related to ENSO and correcting for the recovery from the Mt Pinatubo eruption, our GSFC acceleration is $0.097 \pm 0.050 \text{ mm yr}^{-2}$ (90% CL), consistent with the Nerem et al. (2018) of $0.084 \pm 0.025 \text{ mm yr}^{-2}$ (one standard deviation) over 1993-2017 using the same corrections. Note the projections include the effects of volcanic eruptions prior to 2005, so the observed accelerations should not be corrected for the impacts from the Mt Pinatubo eruption when comparing directly with observations.

Line 186: I suspect that Fig.3 is related to ORA S5. Add this info line 186 as well as the analysis time span.

Agreed. Line 205 revised by adding “over 1970-2017 based on the ORAS5 reanalysis” in the revision.

- Lines 204-208: I suggest you provide deeper discussion about the causes of non uniform acceleration (Fig.3g). The explanation provided here relates the Southern Ocean acceleration to recent warming. Is there a consensus on that? What about other regions like North Pacific, Western Indian Ocean?

Dangendorf et al. (2019) also found notable acceleration in the Southern Ocean over 1968-2015 (their Fig. 3b) based on sea level reconstructions, which is related to the increased ocean heat uptake here caused by the intensification and they state a basin-scale equatorward shift of Southern Hemispheric westerlies. The discussion about acceleration other regions can be found in Dangendorf et al. (2019).

The discussion of regional accelerations is beyond the scope of what we can cover in this manuscript. Unlike Dangendorf et al. (2019) paper, the main objective of this paper is to evaluate the sea-level projections by comparing with available observations, so we try to minimize the natural variability in observation via MVLr model. The target of Fig. 3 is to show regional sea level includes considerable natural variability and the difficulty in isolating the trend and acceleration in the tide-gauge observations. This variability is not directly related to the

comparison with the sea-level projections. Furthermore, the acceleration estimated by ORAS5 is only one component (e.g., not including Antarctic Ice Sheet contributions) of the acceleration impacting the sparse tide-gauge network. Evaluating the ORAS5 reanalysis is not our focus. We addressed this clearly in lines 540-543.

-Lines 212-229: Why not correct for GPS-based VLM where available?

Good comment. In our various sensitivity test we did attempt to use GPS observations. As stated on lines 673-675, we found little difference when comparing our estimated VLM rates with GPS ULR6A solution (<https://www.sonel.org/-ULR6a-.html>). However, the GPS results were noisier and only available at a subset of 103 stations. Hence, here we use the trend of the difference between tide gauge and nearby altimeter measurements at each tide-gauge station to infer local VLM⁷.

-Line 241 (and methods): Did you check whether nearby tide gauge show the same dependency on climate modes?

Yes, this is shown in Fig. 3. For example, tide gauges in the west tropical Pacific Ocean are all show negative regression coefficients, which is dominated by ENSO and PDO indexes (Fig. 3a-b).

- Lines 284-289: Here again, some discussion about the causes of regional variability in acceleration would be useful.

Again, detailed discussion of regional accelerations is beyond the scope of what we can cover in this manuscript.

As we explained in lines 101-109, ‘... , the procedure of multiple-model ensemble averaging in the production of the IPCC AR5 (as well as SROCC) projections reduces but does not completely eliminate the natural variability because of the limited ensemble size used for the AR5 projections (the RCP2.6 contains 16 models, and the RCP4.5 and 8.5 are based on 21 models)²¹. Therefore, we also use low-pass filtered projections developed by CSIRO^{22,23}, in which the dynamic sea level (DSL), the only component directly simulated by the Coupled Model Intercomparison Project Phase 5 (CMIP5) models, is smoothed using a 20-year running-mean filter before adding to other sea-level contributions and computing the ensemble average.’ Hence, the regional variability in acceleration in the AR5 and SROCC is largely related to the remaining natural variability in the multiple-model ensemble, although a regional acceleration signal does emerge on longer time scales. However, ‘the AR5_lp projections produce a more uniform large-scale acceleration than the AR5 and SROCC projections because of reduced natural variabilities (Supplementary Fig. 16). As a result, the AR5_lp sea-level projections under different scenarios can be distinguished by acceleration in the earlier decades of the 21st century than AR5 and SROCC projections’ (lines 305-309).

- Discussion section: As I mentioned above, the fact that trend and acceleration in projections (up to 2032) agree with past observations just mean to me that projections are rather conservative in terms of future contribution from the ice sheets. This is certainly the best that can be done today but this does not preclude from future surprises in the real world. Such comparisons should be regularly reproduced.

We agree that this is the best that can be done at this stage. Thank you.

We also concur that the agreement between the historical observations and the near term projections does not guarantee there will be no rapid acceleration in ice sheet contributions later in the 20th century, and indeed we referred to this possibility (lines 36-37 and lines 358-361) of the previous manuscript (lines 32-37). However, recent publications (Edwards et al. 2019; Golledge et al. 2019) of ice sheet contributions included in the IPCC SROCC projections result in only modestly higher projections than the AR5 assessment.

-Table 1: The date for 'Altimeter GMSL' is wrong (not 1993-2018 but 2007-2018; over 1993-2018, the altimeter rate is only 3.2 mm/yr).

The date in the original Table 1 showed the data lengths and the period used to apply the MVLRL model. However, the actual trends are for 2007-2018. We revised Table 1 to remove what was clearly a number of misleading lines.

Reviewer comments, second round -

Reviewer #4 (Remarks to the Author):

As Reviewer 4, I think that all my questions and comments have been correctly answered and accounted for in this revised manuscript. Overall this new version reads well and is convincing enough to be published in its present form.

Reviewer #5 (Remarks to the Author):

I will focus in this evaluation on the responses of the authors to the points raised by reviewer 3 to the version 244675_1_art_file_4850452_qf7j3s_convrt.pdf

With one exception, I think that the authors have responded well to most of the criticism raised by this reviewer. In particular, some this criticism was due to a misunderstanding of the periods in which the trends were estimated. However, I think that reviewer #3 is correct when they point to a superficial, if not incorrect, statistical analysis, as I explain below.

I have in addition one suggestion that the authors may want to consider. It is a general comment on the definition of acceleration, which I think it is also partly explains in part the criticism/misunderstanding by reviewer #3. My other, more substantive, concern is indeed on the statistical analysis and the estimation of the confidence interval of the estimated trends. In my view, this statistical analysis is too simplistic, perhaps flawed, and it may affect the conclusions of the study. Since the study heavily relies on the comparison of trends, I think that the statistical analysis should accord with the state-of-the-art. It may happen that the application of more correct methods yields in the end the same results, but this should be carefully checked.

1) Definition of acceleration. The authors have assumed that the acceleration can be defined as a quadratic dependency of sea level on time. This is, however, only one possibility. which assumes a very precise form of this temporal dependency. There are other definitions that may be more flexible, and which may also yield different results. For instance, another definition is a monotonic increase in $\Delta SL = SL(t+1) - SL(t)$, where $SL(t)$ is the value of sea level at year (t) . One can assume a linear or other functional form for this monotonic increase. This definition actually would more closely comply with our intuitive understanding of acceleration, as an increase in the annual changes of mean sea level. This definition is what reviewer 3 applied when they claimed that the results show no acceleration, as the estimated trends did not show any change two different periods.

This is just a general comment which may not affect the overall results, as the definition adopted by the authors may also be reasonable and defensible. However, a clarification somewhere in the manuscript that this is an option among others could avoid misunderstandings.

2) My more substantive criticism is on the section Methods (statistical analysis). Here, I think the authors have mixed two different issues and that the authors have not applied the correct methods to estimate the trends and their uncertainties. The authors have applied a t-test in which the number of degrees of freedom is adjusted according to the autocorrelation of the sea level series. This would be a possible approach, although not completely correct, to test for the presence of trends. It is, however, not correct to estimate the uncertainties in the estimated trends and also not correct to test whether two estimated trends are different.

To illustrate my point, the authors may want to consider a time series like this one

$$y(t) = \alpha t + \epsilon(t)$$

where $\alpha = 10$ mm/year and the standard deviation of the residuals ϵ is very small, say

0.1 mm.

This is a very steep trendy series with a little bit of noise added on top of it. The series $y(t)$ is strongly autocorrelated, yet the uncertainty in the estimation of the trend is very small. If we increase α to 100 mm/year, $y(t)$ becomes even more strongly autocorrelated, but the uncertainty in the estimation of the trend obviously should become even smaller. This is at odds with the approach applied by the authors.

The misunderstanding arises actually by the fact that the important aspect in the estimation of the the uncertainties is the autocorrelation of the residuals $\epsilon(t)$, not of the time series $y(t)$. The autocorrelation of the $y(t)$ series is important when trying to test for the presence of a trend. For that type of test, a null-hypothesis needs to be assumed, for instance the $y(t)$ has zero trend but is a realization of an autoregressive process. This is, however, not the actual problem the authors need to solve here. What they are trying to estimate is whether or not two series that are assumed to be realizations of the same trendy statistical model differ in their model parameters (slope).

The problem of estimating the uncertainty in the trend (and in general the estimation of a regression parameter when the residuals are autocorrelated) has a long history, and it is not straightforward to solve. The classical method applied by the authors, which is based on the regression parameter or the slope being t -distributed relies on three assumptions, and one of them is that the residuals $\epsilon(t)$ are temporally independent and normally distributed (see e.g. the paper by Mudelsee cited below). If this assumption is not fulfilled, the uncertainty bounds derived from the t -distribution can be wrong, and quite widely so.

The main problem I see in the methods section is that the present study does not analyse the structure of the residuals at all, and my impression is that for sea level the residuals can indeed be strongly autocorrelated. Even more importantly, if these residuals are shown to display a trend (e.g. they are not temporary stationary by applying a trend test), then it is clear that the statistical model is misspecified. The statistical analysis must then look into the structure of the residuals with the appropriate methods: a non-parametric test of non-stationarity, e.g. Mann-Kendall test, and test its autocorrelation in the context of linear regression, usually the Durbin-Watson test. If the authors applied the DW test to their residuals and this results in no indication of autocorrelation, then the result of the study are likely correct and very little needs to be changed. If not, more complex methods need to be applied to properly estimate the confidence intervals, and the conclusions may need to be changed.

There are several standard methods to be applied in cases with autocorrelated residuals. One is the Cochrane-Orcutt method, for which there is a wikipedia entry broadly describing the problem and the method. I mention the wiki entry to highlight that this is not anything fancy, just a standard approach in statistics. There are other more 'sophisticated' methods based on bootstrapping. The simplest of those would be to estimate a slope α_{est} by the usual Ordinary Least Squares method, obtain a realization of the residuals, generate 1000 realizations of residuals with methods that exactly replicate their autocorrelation function (phase-randomized Fourier Transform, see Ebisuzaki), add the generated residuals to the term α_{est} to create 1000 surrogate time series and estimate again the slope for each of the 1000 bootstrap series. The 5%-95% interval of the bootstrap samples is an estimation of the confidence intervals, and the median would be the central estimate. This bootstrap approach has the advantage that it preserves the distribution and the autocorrelation structure of the residuals. There are other bootstrap approaches, however.

Here are some references that describe those methods:

https://en.wikipedia.org/wiki/Cochrane%E2%80%93Orcutt_estimation

Thejll, Peter, and Torben Schmith. "Limitations on regression analysis due to serially correlated residuals: Application to climate reconstruction from proxies." *Journal of Geophysical Research: Atmospheres* 110.D18 (2005).

Sun, Hongguang, and Sastry G. Pantula. "Testing for trends in correlated data." *Statistics & probability letters* 41.1 (1999): 87-95.

Ebisuzaki, Wesley. "A method to estimate the statistical significance of a correlation when the data

are serially correlated." *Journal of Climate* 10.9 (1997): 2147-2153.

The following paper presents an overview of the width of the problem and offers some possible solutions.

Mudelsee, Manfred. "Trend analysis of climate time series: A review of methods." *Earth-Science Reviews* 190 (2019): 310-322.

A paper with a variety definitions of sea level acceleration
Hünicke, Birgit, and Eduardo Zorita. "Statistical analysis of the acceleration of Baltic mean sea-level rise, 1900–2012." *Frontiers in Marine Science* 3 (2016): 125.

I am sorry to raise new technical issues after the round of reviews. I know it may seem unfair, but hope these comments may be of some help to improve the manuscript.

Reply to Reviewers

We sincerely thank the two reviewers for their constructive and insightful comments. In the detailed reply to the reviewers, the original comments are in black *italics* and our responses are in blue.

Reply to the reviewers

Reply to reviewer #4:

Reviewer #4 (Remarks to the Author):

As Reviewer 4, I think that all my questions and comments have been correctly answered and accounted for in this revised manuscript. Overall this new version reads well and is convincing enough to be published in its present form.

Thank you for your positive comments and encouragement.

Reply to reviewer #5

I will focus in this evaluation on the responses of the authors to the points raised by reviewer 3 to the version 244675_1_art_file_4850452_qf7j3s_convrt.pdf

With one exception, I think that the authors have responded well to most of the criticism raised by this reviewer. In particular, some this criticism was due to a misunderstanding of the periods in which the trends were estimated. However, I think that reviewer #3 is correct when they point to a superficial, if not incorrect, statistical analysis, as I explain below.

I have in addition one suggestion that the authors may want to consider. It is a general comment on the definition of acceleration, which I think it is also partly explains in part the criticism/misunderstanding by reviewer #3. My other, more substantive, concern is indeed on the statistical analysis and the estimation of the confidence interval of the estimated trends. In my view, this statistical analysis is too simplistic, perhaps flawed, and it may affect the conclusions of the study. Since the study heavily relies on the comparison of trends, I think that the statistical analysis should accord with the state-of-the-art. It may happen that the application of more correct methods yields in the end the same results, but this should be carefully checked.

Thank you. These two comments are both valuable and have been responded to. Please see below for detailed responses.

1) Definition of acceleration. The authors have assumed that the acceleration can be defined as a quadratic dependency of sea level on time. This is, however, only one possibility. which assumes a very precise form of this temporal dependency. There are other definitions that may be more flexible, and which may also yield different results. For instance, another definition is a monotonic increases in $\Delta SL = SL(t+1)-SL(t)$, where $SL(t)$ is the value of sea level at year (t). One can assume a linear or other functional form for this monotonic increase. This definition actually would more closely comply with our intuitive understanding of acceleration, as an increase in the annual changes of mean sea level. This definition is what reviewer 3 applied when they claimed that the results show no acceleration, as the estimated trends did not show any change two different periods.

This is just a general comment which may not affect the overall results, as the definition adopted by the authors may also be reasonable and defensible. However, a clarification somewhere in the manuscript that this is an option among others could avoid misunderstandings.

Thank you. We agree there are other acceleration definitions and the quadratic term method we used is one of these methods. We add comments in manuscript to clarify this in Methods part (Lines 595-599): ‘Although there are also other definitions of acceleration allowing us to detect the change of acceleration during the research period (Hünicke et al., 2016), the quadratic term estimated by the MVLRL model is robust estimation of the acceleration by using all the data and simultaneously allowing minimization of the natural variability signals’.

This addition is based on the following investigation of the robustness of the acceleration estimates from three different methods following Hünicke et al. (2016):

(1) Fit to a polynomial of order two in time (method we used in manuscript):

$$y(t) = \alpha + \beta_1 t + \beta_2 t^2 + \varepsilon(t) \quad (1)$$

(2) Calculate the long-term linear trend in the rates computed over gliding 11-year time segments.

(3) Calculate the long-term linear trend of the annual increments of sea level.

Firstly, we use these three methods to estimate the acceleration of the GMSL over 2007-2100 from IPCC AR5 projections under RCP2.6, 4.5 and 8.5 scenarios. The accelerations estimated by different methods are almost the same (Figure A1, Table A1), supporting a robust estimation for GMSL in which the regional variability is much reduced.

Figure A1. The acceleration estimate using three different methods. (a) Solid curves present the GMSL from AR5 projections under three RCP scenarios. The dashed lines are the linear-plus-quadratic fit estimated based on method 1. (b) Solid curves are the 11-year running trend under three RCP scenarios based on method 2, and the dashed lines present the linear trend of these running trend time series over the whole period. (c) Solid curves denote the annual increments of GMSL under three RCP scenarios based on method 3, and the dashed lines are the linear trend of the annual increment time series over 2007-2100. (Note the anomaly in the 2050s is a known issue with the AR5 projections and relates to some unknown glitch in the original model projections (probably from just one model)).

Table A1. Accelerations of AR5 GMSL under three RCP scenarios over 2007-2100, using three estimation methods.

	OLS	RT	AINC
RCP 2.6	0.002	0.004	0.002
RCP 4.5	0.031	0.029	0.030
RCP 8.5	0.098	0.096	0.097

OLS, fit to a second order time polynomial (Method 1); RT, running linear trends with ordinary linear regression (Method 2); AINC, annual increments and ordinary linear regression (Method 3). Unit is mm year⁻².

On a regional scale, we select one typical tide gauge station located at the New York City to test the robustness of these three methods (Figure A2). The results from the method 1 agree well with acceleration values from the method 2 (Table A2). The accelerations based on method 3 show larger discrepancy when compared with the other two methods. This difference between method 3 and method 1 or 2 may be related to the regional variability in the sea-level time series.

Another potential reason is the natural variability may still remain in the AR5 projections, since the procedure of multiple-model ensemble averaging in the production of the IPCC AR5 projections reduces but does not completely eliminate the natural variability because of the limited ensemble size used for the AR5 projections (Lyu, et al., 2015). Both the regional variability and the residual natural variability could lead to the large noise in the annual increment time series then bias the acceleration estimate from method 3 (Figure A2c).

Figure A2. The same as Figure A1, but for AR5 projection located at the New York tide gauge station.

Table A2. The same as Table A1, but for AR5 projection located at the New York tide gauge station.

	OLS	RT	AINC
RCP 2.6	-0.024	-0.017	-0.017
RCP 4.5	0.021	0.024	0.040
RCP 8.5	0.102	0.105	0.081

Therefore, we also use the AR5_lp projections, in which the dynamic sea level (DSL) is smoothed using a 20-year running-mean filter before adding to other sea-level contributions and computing the ensemble average. The AR5_lp projections are very similar to the AR5 projections, but the inter-annual to decadal natural variability has been much reduced, which helps us to focus on the climate change signals (Zhang et al., 2017). To further test the impact from natural variability on the method 3, we apply these three methods to the AR5_lp projections (Figure A3). The acceleration estimates of AR5_lp projections using these three methods become consistent again on regional scale (Table A3), and the variability in the annual increment time series is largely reduced (Figure A3c) compared with that from AR5 projections (Figure A2c). Hence, the acceleration estimated based on method 3 is more sensitive to the natural variability in a time series. In fact, the 11-year running trend time series of AR5 projection based on method 2 also present large variability signal (e.g., the low-frequency oscillation before 2040 under RCP 8.5 scenario in Figure A2b), while the acceleration estimate does not change much compared with that from AR5_lp projection due to the long time period. Furthermore, we also use the satellite altimeter observations (1993-2018; 26 years), and the 11-year running trend process used in method 2 will loss about a half records for altimeter observations at the beginning and ending. Hence, method 2 is not appropriate for a relatively short study period. Although methods 2 and 3 allow us to detect the change of acceleration during a research period, they should be used over a long research period over which the signal-to-noise ratio is high. In fact, method 1 used in our paper is a robust method to estimate acceleration which uses all the data simultaneously and allow minimization of the natural variability signals as well, so we stick with our original method 1. In the updated manuscripts, we stress method 1 is one of the acceleration definition methods (Lines 595-599).

Figure A3. The same as Figure A1, but for AR5_lp projection located at the New York tide gauge station.

Table A3. The same as Table A1, but for AR5_lp projection located at the New York tide gauge station.

	OLS	RT	AINC
RCP 2.6	-0.027	-0.026	-0.024
RCP 4.5	0.012	0.012	0.013
RCP 8.5	0.079	0.078	0.078

2) My more substantive criticism is on the section *Methods (statistical analysis)*. Here, I think the authors have mixed two different issues and that the authors have not applied the correct methods to estimate the trends and their uncertainties. The authors have applied a t-test in which the number of degrees of freedom is adjusted according to the autocorrelation of the sea level series. This would be a possible approach, although not completely correct, to test for the presence of trends. It is, however, not correct to estimate the uncertainties in the estimated trends and also not correct to test whether two estimated trends are different.

To illustrate my point, the authors may want to consider a time series like this one

$$y(t) = \alpha t + \epsilon(t)$$

where $\alpha = 10$ mm/year and the standard deviation of the residuals $\epsilon(t)$ is very small, say 0.1 mm.

This is a very steep trendy series with a little bit of noise added on top of it. The series $y(t)$ is strongly autocorrelated, yet the uncertainty in the estimation of the trend is very small. If we increase α to 100 mm/year, $y(t)$ becomes even more strongly autocorrelated, but the uncertainty in the estimation of the trend obviously should become even smaller. This is at odds with the approach applied by the authors.

The misunderstanding arises actually by the fact that the important aspect in the estimation of the the uncertainties is the autocorrelation of the residuals $\epsilon(t)$, not of the time series $y(t)$. The autocorrelation of the $y(t)$ series is important when trying to test for the presence of a trend. For that type of test, a null-hypothesis needs to be assumed, for instance the $y(t)$ has zero trend but is a realization of an autoregressive process. This is, however, not the actual problem the authors need to solve here. What they are trying to estimate is whether or not two series that are assumed to be realizations of the same trendy statistical model differ in their model parameters (slope).

The problem of estimating the uncertainty in the trend (and in general the estimation of a regression parameter when the residuals are autocorrelated) has a long history, and it is not straightforward to solve. The classical method applied by the authors, which is based on the regression parameter or the slope being t-distributed relies on three assumptions, and one of them is that the residuals $\epsilon(t)$ are temporally independent and normally distributed (see e.g. the paper by Mudelsee cited below). If this assumption is not fulfilled, the uncertainty bounds derived from the t-distribution can be wrong, and quite widely so.

The main problem I see in the methods section is that the present study does not analyse the structure of the residuals at all, and my impression is that for sea level the residuals can indeed be strongly autocorrelated. Even more importantly, if these residuals are shown to display a trend (e.g. they are not temporary stationary by applying a trend test), then it is clear that the

statistical model is misspecified. The statistical analysis must then look into the structure of the residuals with the appropriate methods: a non-parametric test of non-stationarity, e.g. Mann-Kendall test, and test its autocorrelation in the context of linear regression, usually the Durbin-Watson test. If the authors applied the DW test to their residuals and this results in no indication of autocorrelation, then the result of the study are likely correct and very little needs to be changed. If not, more complex methods need to be applied to properly estimate the confidence intervals, and the conclusions may need to be changed.

There are several standard methods to be applied in cases with autocorrelated residuals. One is the Cochrane-Orcutt method, for which there is a wikipedia entry broadly describing the problem and the method. I mention the wiki entry to highlight that this is not anything fancy, just a standard approach in statistics. There are other more 'sophisticated' methods based on bootstrapping. The simplest of those would be to estimate a slope α_{est} by the usual Ordinary Least Squares method, obtain a realization of the residuals, generate 1000 realizations of residuals with methods that exactly replicate their autocorrelation function (phase-randomized Fourier Transform, see Ebisuzaki), add the generated residuals to the term α_{est} to create 1000 surrogate time series and estimate again the slope for each of the 1000 bootstrap series. The 5%-95% interval of the bootstrap samples is an estimation of the confidence intervals, and the median would be the central estimate. This bootstrap approach has the advantage that it preserves the distribution and the autocorrelation structure of the residuals. There are other bootstrap approaches, however.

Thanks for your constructive suggestions and detailed guidance. We followed your suggestions and applied the Durbin-Watson test on both the observations (satellite altimeter and tide gauges) and sea level projection, we found there is indeed significant auto-correlation in the residuals. Here we try both the Cochrane-Orcutt (CO) and bootstrapping methods and compare with our results. As shown below, the main conclusions do not change. In the revised manuscript, we follow your suggestions and change our statistic method to bootstrapping and emphasize the auto-correlation in the residual.

1. CO method

Followed Thejll et al. (2005), the CO method processes are:

Step 1: For a multiple variable regression model:

$$y_t = \alpha + \sum_{k=1}^K \beta_k x_t^{(k)} + u_t \quad (2)$$

where the $x_t^{(1)} = time$, $x_t^{(2)} = time^2$, $x_t^{(3)} = ENSO$, $x_t^{(4)} = PDO$,

run ordinary least square (OLS) regression and get the residual time series (u_t).

Step 2: Assume u_t following an AR(1) process with the auto-correlation at lag 1 being ρ :

$$u_t = \rho u_{t-1} + \varepsilon_t \quad (3)$$

where ε_t are typically assumed to be uncorrelated random variables. Find an estimate for ρ using OLS regression.

Step 3: Rewrite Eq. (2) at time step $t-1$:

$$y_{t-1} = \alpha + \sum_{k=1}^K \beta_k x_{t-1}^{(k)} + u_{t-1} \quad (4)$$

Step 4: calculate Eq. (2) - Eq. (4) $\times \rho$, rearrange the term in combination with (2):

$$y_t - \rho y_{t-1} = \alpha(1 - \rho) + \sum_{k=1}^K \beta_k (x_t^{(k)} - \rho x_{t-1}^{(k)}) + \varepsilon_t \quad \text{for } t = 2, 3, \dots, N \quad (5)$$

where ε_t satisfy the no autocorrelation assumption for OLS.

Step 5: run OLS regression on Eq. (5), new α^* , β_k^* , and u_t^* can be estimated, then calculate the ρ^* based on u_t^* (Step 2), repeat Step 2 ~ Step 4 until the estimated value of ρ is converged:

$$\text{abs}\left(\frac{\tilde{\rho} - \rho^*}{\rho^*}\right) \leq \delta, \quad (\delta = 10^{-6}) \quad (6)$$

Step 6: When Eq. (6) is satisfied, the estimators are the corresponding $\tilde{\alpha}$, $\tilde{\beta}_k$, The uncertainties are estimated from the variance of the residuals.

Here we take GMSL from NASA/GSFC as an example to show the details. Consider a multiple variable linear regression model including the ENSO and PDO indexes:

$$y(t) = \alpha + \beta_1 t + \beta_2 t^2 + \beta_3 ENSO(t) + \beta_4 PDO(t) + u(t) \quad (7)$$

Firstly, the Durbin-Watson test indicates rejection of the null hypothesis at the 10% significance level, in favour of the alternative hypothesis that the autocorrelation among residuals is greater than zero. The residual time series $u(t)$ exhibit a lag-1 auto-correlation coefficient of 0.61.

Then we use the CO method and the iteration gets converged after 6 steps, with the lag-1 auto-correlation coefficient reduces to -0.06 (Figure A4b). The estimated regression coefficients based on CO method do not change much compared with the values from the OLS method (Table A4).

Figure A4. Comparison between the OLS and CO methods. (a) GMSL from satellite observation (blue), predicted GMSL fit based on OLS method (red) and CO method (green), respectively. (b) Residual time series from OLS method (red) and CO method (green). The values in brackets denote the lag-1 auto-correlation coefficient of the residual time series.

Table A4. The estimated regression coefficients with confidence level (90%) based on OLS and CO methods.

	β_1	β_2	β_3	β_4
OLS	3.397 ± 0.224	0.025 ± 0.027	1.947 ± 1.056	1.3474 ± 0.983
CO	3.390 ± 0.225	0.026 ± 0.027	1.945 ± 1.056	1.1165 ± 0.987

We also apply the CO method to the other GMSL observations and 177 tide gauge stations to repeat our main results in the manuscript (Table 1 is copied below for easier comparison).

Generally, the updated results based on CO methods change very little compared with the OLS values (Table A5), with a slightly increased uncertainty range after using the CO method. This comparable uncertainty may be because we consider the autocorrelation by using the effective degree of freedom (one degree of freedom for about every four original monthly data points) when estimated the uncertainty in our manuscript.

However, the CO method sometimes cannot solve the auto-correlated residual well. Take the TG ID 48 from AR5_lp projection under RCP2.6 as example (Figure A5), this station cannot get convergence even after 100 iterations, with residual autocorrelation coefficient of 0.49 at the 100th iteration step. This may be related to the AR(1) process assumption for the residual time series, so cannot solve the auto-correlated residual on low-frequency well (Mudelsee, 2019). Note that the size of the residuals is small – the order of 1% of the signal.

Figure A5. An example tide-gauge station where the CO method doesn't perform well. (a) The original time series from AR5_lp projections under RCP 2.6 located in the nearest grid point from the TG ID 48 (tide-gauge locations please see Fig. 2 in manuscript) in black, the linear-plus-quadratic fit estimated by the OLS is in red. (b) The residual time series from the OLS estimate.

Table 1 Global mean and regional weighted mean sea-level trends (mm yr⁻¹) over the common period (2007-2018) and accelerations (mm yr⁻²) during the whole study period of each data. Altimeter and reconstructed GMSL have the climate variability related to ENSO and PDO removed over the whole period of the records. The uncertainties represent the 90% CL.

	Trend	Acceleration
Altimeter GMSL	2007-2018	1993-2018
GIA-adjust	3.9 ± 0.5	0.084 ± 0.060
GPS-adjust	3.9 ± 0.5	0.086 ± 0.061
NASA/GSFC	3.8 ± 0.5	0.051 ± 0.051
GMSL reconstruction	2007-2015	1970-2015
D2019	3.7 ± 0.3	0.063 ± 0.011
GMSL reconstruction	2007-2018	1970-2018
CW2011	3.7 ± 0.5	0.062 ± 0.026
AR5 projections	2007-2018	2007-2032
RCP2.6	3.9 ± 0.6	0.035 ± 0.049
RCP4.5	3.8 ± 0.5	0.048 ± 0.047
RCP8.5	3.9 ± 0.6	0.069 ± 0.048
SROCC projections	2007-2018	2007-2032
RCP2.6	3.8 ± 0.6	0.033 ± 0.051
RCP4.5	3.7 ± 0.6	0.051 ± 0.050
RCP8.5	3.9 ± 0.6	0.082 ± 0.054
TG weighted-mean at TG locations	2007-2018	1970-2018
TG	3.6 ± 1.8	0.066 ± 0.095
AR5 regional weighted-mean at TG locations	2007-2018	2007-2032

RCP2.6	4.0 ± 1.3	0.021 ± 0.084
RCP4.5	3.7 ± 0.9	0.053 ± 0.060
RCP8.5	3.8 ± 0.8	0.074 ± 0.088
		
SROCC regional weighted mean at TG locations	2007-2018	2007-2032
RCP2.6	3.9 ± 1.3	0.021 ± 0.085
RCP4.5	3.5 ± 0.9	0.057 ± 0.062
RCP8.5	3.8 ± 0.8	0.089 ± 0.087
		
AR5_lp regional weighted- mean at TG locations	2007-2018	2007-2032
RCP2.6	4.1 ± 1.3	0.041 ± 0.022
RCP4.5	3.9 ± 1.3	0.053 ± 0.018
RCP8.5	4.1 ± 1.3	0.071 ± 0.024

Table A5. The same as Table 1, but based on the CO method.

	Trend	Acceleration
Altimeter GMSL	2007-2018	1993-2018
GIA-adjust	4.1 ± 0.5	0.083 ± 0.069
GPS-adjust	4.1 ± 0.5	0.093 ± 0.069
NASA/GSFC	3.8 ± 0.5	0.070 ± 0.052
GMSL reconstruction	2007-2015	1970-2015
D2019	3.7 ± 0.3	0.062 ± 0.012
GMSL reconstruction	2007-2018	1970-2018
CW2011	3.7 ± 0.5	0.064 ± 0.027
AR5 projections	2007-2018	2007-2032
RCP2.6	4.0 ± 0.6	0.025 ± 0.049
RCP4.5	3.9 ± 0.6	0.031 ± 0.051
RCP8.5	3.7 ± 0.5	0.052 ± 0.047
SROCC projections	2007-2018	2007-2032
RCP2.6	3.8 ± 0.6	0.033 ± 0.051
RCP4.5	3.7 ± 0.6	0.051 ± 0.050
RCP8.5	3.9 ± 0.6	0.082 ± 0.054
TG weighted-mean at TG locations	2007-2018	1970-2018
TG	3.6 ± 1.8	0.055 ± 0.124
AR5 regional weighted-mean at TG locations	2007-2018	2007-2032
RCP2.6	4.0 ± 1.5	0.015 ± 0.101
RCP4.5	3.7 ± 0.9	0.055 ± 0.072
RCP8.5	3.8 ± 0.8	0.077 ± 0.111

SROCC regional weighted mean at TG locations	2007-2018	2007-2032
RCP2.6	3.9 ± 1.5	0.021 ± 0.101
RCP4.5	3.5 ± 0.9	0.057 ± 0.072
RCP8.5	3.8 ± 0.8	0.089 ± 0.111

AR5_lp regional weighted-mean at TG locations	2007-2018	2007-2032
RCP2.6	4.2 ± 1.4	0.021 ± 0.028
RCP4.5	3.9 ± 1.3	0.053 ± 0.020
RCP8.5	4.1 ± 1.3	0.066 ± 0.027

2. Bootstrapping method

We also take GMSL from NASA/GSFC as an example to show the detailed procedure of bootstrapping method. Followed Ebisuzaki (1997), the bootstrapping method procedure are:

Step 1: Consider a multiple variable linear regression model including the ENSO and PDO indexes:

$$y_i = \alpha + \beta_1 t_i + \beta_2 t_i^2 + \beta_3 ENSO_i + \beta_4 PDO_i + \varepsilon_i = \hat{y}_i + \varepsilon_i \quad (8)$$

where $i = 1, 2, \dots, N$, and N is the length of the time series. ε_i presents the residual time series, \hat{y}_i is the predicted fit.

Step 2: Generate 1,000 artificial residual time series by using the phase-randomized sampling procedure (Ebisuzaki, 1997). This resampling procedure preserves the power spectrum, so the resampled series retains the same autocorrelation function as the original residual series (Figure A6).

Step 3: Add the 1000 artificial residuals (Figure A7a) back to the estimators to generate 1000 surrogate time series (Figure A7b):

$$y_i^k = \hat{y}_i + \varepsilon_i^k, \quad (k = 1, 2, \dots, 1000; i = 1, 2, \dots, N) \quad (9)$$

Step 4: Linear trend (acceleration) of each surrogate time series is estimated, with a pool of 1,000 derived trends (accelerations).

Step 5: The central value of trend (acceleration) is determined from the median of 1000 derived trends (accelerations), and the associated uncertainty is the STD of 1000 trends (accelerations). The 90% CL is computed as the final uncertainty (i.e., STD) multiplied by 1.65 assuming a normal distribution.

For the GMSL from NASA/GSFC, the central values based on bootstrapping method agree well with results based on the OLS method, while the uncertainty of the estimator tend to become slightly smaller (Table A6).

Figure A6. Comparison between residual time series (ϵ_i) estimated from the MVLR model in Eq. (8) based on NASA/GSFC observations and its 1,000 resampling residuals. (a) The single-sided amplitude spectrums from the original residuals (ϵ_i) ; black) and 1,000 resampling residuals (r_i^k) ; grey). (b) The autocorrelation functions from the original residuals (ϵ_i) ; black) and 1,000 resampling residuals (r_i^k) ; grey).

Figure A7. (a) The residual time series (ϵ_i) estimated from the MVLR model in Eq. (8) based on NASA/GSFC observations (black) and its 1,000 resampling residuals (gray). (b) The GMSL from NASA/GSFC (black) and its 1,000 surrogate time series (gray).

Table A6. The estimated regression coefficients with confidence level (90%) based on OLS and bootstrapping methods.

	β_1	β_2	β_3	β_4
OLS	3.4 ± 0.2	0.025 ± 0.027	1.95 ± 1.06	1.35 ± 0.99
Bootstrapping	3.0 ± 0.1	0.026 ± 0.013	1.89 ± 0.47	0.88 ± 1.04

We also apply the bootstrapping method to the other GMSL observations and 177 tide gauge stations to repeat our main results. For the central values of trend on both global mean and regional scales, the difference between OLS and bootstrapping methods are smaller than 0.1

mm yr⁻¹ (the second columns in Table A7 and Table 1), with the difference in uncertainty range also smaller than 0.1 mm yr⁻¹. The changes in central values of acceleration are smaller than 0.004 mm yr⁻² (the third columns in Table A7 and Table 1) after using the bootstrapping methods. The estimated uncertainty based on the bootstrapping methods also agrees well with the OLS, except the uncertainty of acceleration from NASA/GSFC becomes smaller and significant, and the acceleration uncertainty from the TG regional weighted-mean result becomes slightly larger. However, these small differences will not change our main conclusions.

Table A7. The same as Table 1, but based on the bootstrapping method.

	Trend	Acceleration
Altimeter GMSL	2007-2018	1993-2018
GIA-adjust	4.0 ± 0.4	0.083 ± 0.053
GPS-adjust	4.0 ± 0.4	0.089 ± 0.054
NASA/GSFC	3.8 ± 0.3	0.053 ± 0.026
GMSL reconstruction	2007-2015	1970-2015
D2019	3.7 ± 0.3	0.062 ± 0.014
GMSL reconstruction	2007-2018	1970-2018
CW2011	3.7 ± 0.3	0.066 ± 0.011
AR5 projections	2007-2018	2007-2032
RCP2.6	3.9 ± 0.6	0.035 ± 0.052
RCP4.5	3.8 ± 0.6	0.048 ± 0.048
RCP8.5	3.9 ± 0.5	0.067 ± 0.049
SROCC projections	2007-2018	2007-2032
RCP2.6	3.8 ± 0.6	0.036 ± 0.052
RCP4.5	3.7 ± 0.6	0.052 ± 0.050
RCP8.5	3.9 ± 0.6	0.084 ± 0.056
TG weighted-mean at TG locations	2007-2018	1970-2018

TG	3.6 ± 1.7	0.063 ± 0.120
AR5 regional weighted-mean at TG locations	2007-2018	2007-2032
RCP2.6	4.0 ± 1.3	0.021 ± 0.085
RCP4.5	3.7 ± 0.9	0.053 ± 0.063
RCP8.5	3.9 ± 0.8	0.073 ± 0.088
SROCC regional weighted mean at TG locations	2007-2018	2007-2032
RCP2.6	3.9 ± 1.3	0.020 ± 0.085
RCP4.5	3.6 ± 0.9	0.056 ± 0.063
RCP8.5	3.8 ± 0.8	0.089 ± 0.087
AR5_lp regional weighted-mean at TG locations	2007-2018	2007-2032
RCP2.6	4.1 ± 1.3	0.041 ± 0.022
RCP4.5	3.9 ± 1.3	0.053 ± 0.019
RCP8.5	4.1 ± 1.3	0.072 ± 0.024

Here are some references that describe those methods:

https://en.wikipedia.org/wiki/Cochrane%E2%80%93Orcutt_estimation

Thejll, Peter, and Torben Schmith. "Limitations on regression analysis due to serially correlated residuals: Application to climate reconstruction from proxies." Journal of Geophysical Research: Atmospheres 110.D18 (2005).

Sun, Hongguang, and Sastry G. Pantula. "Testing for trends in correlated data." Statistics & probability letters 41.1 (1999): 87-95.

Ebisuzaki, Wesley. "A method to estimate the statistical significance of a correlation when the data are serially correlated." Journal of Climate 10.9 (1997): 2147-2153.

The following paper presents an overview of the width of the problem and offers some possible solutions.

Mudelsee, Manfred. "Trend analysis of climate time series: A review of methods." Earth-Science Reviews 190 (2019): 310-322.

A paper with a variety definitions of sea level acceleration

Hünicke, Birgit, and Eduardo Zorita. "Statistical analysis of the acceleration of Baltic mean sea-level rise, 1900–2012." Frontiers in Marine Science 3 (2016): 125.

I am sorry to raise new technical issues after the round of reviews. I know it may seem unfair, but hope these comments may be of some help to improve the manuscript.

Thanks again for this list of reference, we read these papers carefully. The detailed comment about the statistical methods gives us constructive technical guidance and help us improve the manuscript.

Reviewer comments, third round

Reviewer #5 (Remarks to the Author):

I thank the authors very much for considering my suggestions, which this version fully addresses. I am happy to recommend this version of manuscript for publication.

Reply to Reviewer #5

REVIEWERS' COMMENTS

Reviewer #5 (Remarks to the Author):

I thank the authors very much for considering my suggestions, which this version fully addresses. I am happy to recommend this version of manuscript for publication.

We appreciate your insightful and constructive suggestions, which give us detailed guidance and helped us improve our manuscript on the statistical methods. We sincerely thank you for your positive comments and encouragement.